# GITaR: Generalized Irregular Time Series Regression via Masking and Reconstruction Pretraining

## Abstract

Multivariate time series regression, encompassing forecasting and interpolation, is crucial for numerous real-world applications, particularly in healthcare, climate science, ecology, and others. While recent work has focused on improving modeling for time series regression, two main limitations persist. First, the prevalence of irregularly sampled time series with missing values poses significant challenges. For instance, healthcare applications often involve predicting future or missing observations from irregular data to enable continuous patient monitoring and timely intervention. As current approaches mainly rely on the assumptions of regular time series such as strong periodicity, when applied to irregular ones they exhibit performance degradation. Second, while some state-of-the-art methods (SOTA) do model irregularity and perform regression tasks on irregular data, they are often trained in a fully supervised manner. This limits their ability to generalize easily to different domains (e.g., training and testing datasets with different numbers of variables). To address these challenges, we propose **GITaR**, a **G**eneralized **I**rregular **T**ime Series Regression model via m**a**sking and **R**econstruction pretraining mechanism, aiming to capture the inherent irregularity in time series and learn robust, generalizable representations without supervision for downstream regression tasks. Comprehensive experiments on common real-world regression tasks in healthcare, human activity recognition, and climate science underline the superior performance of GITaR compared to state-of-the-art methods. Our results highlight our model's unique capability to generalize across different domains, demonstrating the potential for broad applicability in various fields requiring accurate temporal prediction and interpolation.

## 1 Introduction

Multivariate time series regression, i.e., regression tasks that encompass interpolation and forecasting, aim to predict the continuous and numerical values based on their relationship within or beyond the existing time range (Tan et al., 2021). This type of regression is particularly challenging when observation data are unevenly sampled and contain missing values, a common issue in many fields. Modeling irregular time series is essential in various real-world applications, including healthcare monitoring, climate science, ecology and more Shukla and Marlin (2021a). For instance, in intensive care units (ICUs), patient data is often collected at varying intervals and from multiple sensors, resulting in irregular time series. These sensors typically monitor physiological features such as heart rate, blood pressure, and respiratory rate. Modeling such irregular times in the past 24 hours to predict the subsequent 24 hours is particularly valuable for continuous patient health condition monitoring and inference.

While advanced deep learning architectures have significantly improved time series analysis, they often rely on the assumption of regular spacing between observations, which limits their realistic applicability to irregular time series data. For example, TS2Vec (Yue et al., 2022) assumes strong auto-correlation in time series, and PatchTST (Wu et al., 2022) initially segment time series into sub-series level patches and assumes observations are continuous without missing. Both methods, however, struggle with irregular time series. As illustrated in Figure 1a, these state-of-the-art (SOTA)

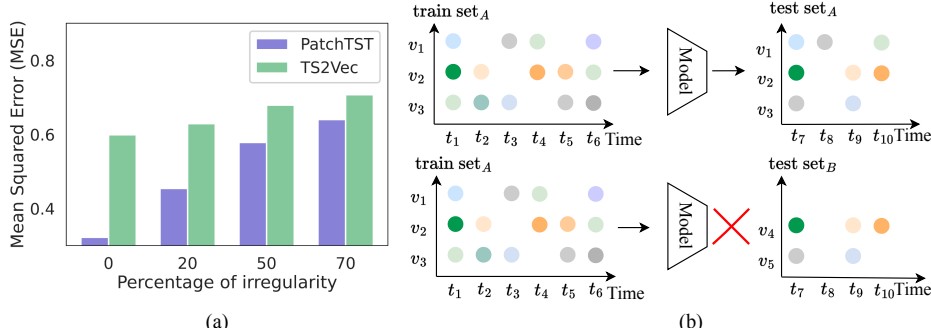

Figure 1: (a) Comparison of MSE between mainstream models on a semi-synthetic irregular dataset with different percentages of irregularity. (b) Irregular time series models cannot be generalized to different domains.

time series modeling methods fail to forecast irregular time series, leading to a significant increase in mean square error (MSE) as irregularity increases.

Attempts have been made to better model irregular time series, primarily under supervised settings. Four predominant approaches have emerged, *set-based models* (Tipirneni and Reddy, 2022a), which include time as an additional feature with values as pairs; *ODE-based models*(Chen et al., 2019), which leverage differential equations to model irregular and continuous time dynamics; *attention-based models*(Shukla and Marlin, 2021b; Zhang et al., 2019; Chen et al., 2024), which extend Transformer (Vaswani et al., 2023) architectures to capture irregularities in temporal dynamics, and *graph-based* (Zhang et al., 2022; 2024a), which address intercorrelations in multivariate irregular time series data. Due to their innovative design for encoding irregularities more effectively, these methods outperform traditional time series models that are designed for regular data patterns. However, these methods still face limitations in robustness and generalizability due to their supervised end-to-end (E2E) training. Typically, these models are trained and tested on the same dataset for a specific task (e.g., training on 2-lead ECG data and testing on the same 2-lead ECG dataset for heart rhythm classification). Consequently, they fail to generalize across different domains, as shown in Figure 1b, including i). varying numbers of variables between training and test sets (e.g., training on 3-lead ECG data but testing on 2-lead ECG data), or ii). different irregularity ratios or missing data patterns between training and test datasets, which can occur due to domain-specific reasons (e.g., ICU data irregularity(test set A) might be caused by the clinician's availability, while missing data in wearable devices for human activity monitoring often results from device detachment (test set B). These limitations hinder current methods from learning universal and generalized representations that are not biased toward specific domains.

Recent advances in self-supervised learning (SSL) have shown promise in modeling time series data without explicit supervision, potentially improving generalization capabilities (Zhang et al., 2024b). However, most SSL approaches still focus on regular time series and cannot be easily adapted to irregular data (Yang et al., 2023; Yue et al., 2022; Zhang et al., 2024b), as we will show in our evaluation. One recent study, Primenet (Chowdhury et al., 2023) targets irregular time series SSL and designs suitable augmentations combined with contrastive learning to learn the characteristics of irregularly sampled data during the pretraining stage. However, its applicability to regression tasks remains less explored. Moreover, PrimeNet still struggles to generalize across different domains due to its modeling of intercorrelations within specific numbers of training variables, limiting its effectiveness beyond the validation datasets used in the study.

Given these challenges and limitations in existing approaches, we aim to address two critical research questions:

- *How can we design a more accurate and robust self-supervised learning framework specifically tailored for irregular time series regression?*

- *How can we develop a generalized SSL modeling approach that can adapt to different domains in irregular time series regression?*

We propose GITaR, an irregular-sensitive reconstruction-based pretraining mechanism to learn more generalizable and robust representations. Among various SSL pretext tasks, masking and reconstruction-based methods have emerged as particularly powerful. These approaches, which primarily focus on reconstructing missing parts of the data, have demonstrated strong predictive capabilities and have been successfully applied in recent large pretrained models (Brown et al., 2020). This approach appears especially suitable for regression tasks, as the underlying principle of masking and reconstruction closely aligns with the interpolation task of predicting missing or future values based on partial data (He et al., 2021; Zhu et al., 2024). Specifically, to leverage the temporal properties of irregular time series, GITaR first introduces an irregular-sensitive masking strategy and time-sensitive patching segmentation. Further, we apply an irregular-temporal encoder that transforms patched segments into latent space representations for learning local and global temporal dynamics. Specifically, within the encoder, the channel-independent design of our approach caters to domains with varying numbers of variables or irregularity ratios, enhancing its generalization capabilities across different domains.

To support the practical evaluation of GITaR, we conduct experiments on 4 naturally occurring irregular and asynchronous time series from healthcare to climate sensing. Our contribution includes:

- We highlight that irregular time series are common and present a significant challenge in real-world applications and find that existing time series modeling algorithms undergo substantial performance degradation when applied to irregular time series regression.

- We propose GITaR, a pioneering pretraining framework uniquely designed for irregular multivariate time series regression. Our approach introduces a novel irregular-sensitive masking and patching technique coupled with an adaptive irregular-temporal encoder, enabling simultaneous learning of local irregularities and global temporal correlations, which is effective for various regression tasks. The results showcase the SOTA performance of GITaR again baselines with an average 5.68% improvement on irregular time series forecasting tasks and an average 6.73% improvement on irregular time series interpolation tasks, demonstrating its effectiveness and robustness to varying irregular patterns.

- GITaR demonstrated remarkable cross-domain generalization capabilities across four diverse real-world datasets, spanning varying numbers of channels, irregular ratios and patterns, temporal resolutions, and domain-specific challenges, establishing GITaR as a versatile foundational model for irregular time series regression. Its adaptability to unseen irregular patterns makes it particularly valuable for real-world applications where data distributions may unexpectedly shift.

## 2 RELATED WORK

**Multivariate Time Series Modeling for Regression Tasks.** Regression tasks such as forecasting or interpolation aim to predict the unseen or missing observation values in time series (Wen et al., 2023; Zhang et al., 2024b). Recent advanced work has shown promising results either trained in a supervised manner or self-supervised manner, such as Informer (Zhou et al., 2021), Crossformer (Zhang and Yan, 2023), PatchTST (Nie et al., 2023) and TS2Vec (Yue et al., 2022). However, these approaches often focus on regular time series forecasting tasks, via modifying the native structure of Transformers to encode temporal dependencies for regular long-term time series forecasting. They neglected the intrinsic characteristics of irregularity and lack of synchronization inherent in multivariate irregular time series data. Therefore, they cannot be readily generalized to multivariate irregular time series analysis.

**Irregular Time Series Modeling.** Irregular time series are characterized by varying time intervals between adjacent observations (Shukla and Marlin, 2021a). Early methods rely on set-based approaches, incorporating the time index as an additional feature and using recurrent networks to learn irregular temporal time dynamics (Che et al., 2016; Schirmer et al., 2022). Another line of work leverages ODEs by parameterizing the governing function in ODEs with neural networks. Additionally, these methods combine ODEs with recurrent structures to learn the underlying dynamics of time series, inherently addressing irregularity (Chen et al., 2019; Rubanova et al., 2019). Attention mechanism in Transformers have also been improved to process irregular time series (Vaswani et al., 2023). For example, mTand (Shukla and Marlin, 2021b) first replaces the positional encoding with fixed continuous time embedding and then maps the irregular input into regular latent

space. To account for the inter-channel dependencies in the multivariate time series, graph-based approaches (Zhang et al., 2022; 2024a) have also been explored by treating each variable as a node in the graph and learning the edge weights to capture the multivariate correlation. However, they are primarily trained under fully or semi-supervised paradigms (Tipirneni and Reddy, 2022b; Zhang et al., 2023), requiring large amounts of high-quality labeled or observed data. In contrast, approaches like PrimeNet (Chowdhury et al., 2023) and PATIS (Beebe-Wang et al., 2023) leverage self-supervised settings to learn representations of irregular time series and improve downstream tasks with limited labeled data. However, current SSL methods remain less explored for regression tasks. Moreover, they employ domain-specific model structures by applying channel-dependent structures directly to the input to learn the correlations between multivariates, potentially hindering their adaptability to diverse datasets with different numbers of variables and tasks.

In contrast, our work seeks to fill these gaps by designing a generalized SSL pretext task for irregular time series regression, not only effectively capturing the intrinsic characteristic in irregular time series but also learn more robust and generalizable representations.

## 3 METHODOLOGY

In this section, we present the details of our proposed GITaR, a masking and reconstruction pretraining mechanism designed to effectively handle irregularities in real-world time series data and learn robust representations that can generalize across various domains.

### 3.1 PRELIMINARIES

A *multivariate irregular time series* is denoted as $\mathbf{O} = (O^1, O^2, ..., O^D)$, where $O^i$ represents the $i^{th}$ variable among $D$ variables. Each $O^i$ is a *univariate irregular time series* represented as $O^i = (o^i_{t_1}, o^i_{t_2}, ..., o^i_{t_{N_i}})$, recorded at time stamp $t_n, n \in [1, N_i]$. Unlike regular time series, the time interval between two consecutive measurements $\Delta t = t_{n+1} - t_n$ is not constant. The number of observations $N_i$ may also vary between different variables. We first synchronize and align the $D$ variables through the upsampling of each $O^i$ to the maximum sample rate, ensuring a consistent number of samples $N$ for each variable. To preserve the irregularity, we simultaneously introduce additional mask variables $M$ and a time index $T$ to account for irregularities. Specifically, each *multivariate irregular time series* can be represented as $\mathbf{O} = (O^1, O^2, ..., O^D) = (\mathbf{T}, \mathbf{X}, \mathbf{M})$ where: $\mathbf{T} \in \mathbb{R}^{N \times D}$ represents the union of timestamps at which any of the $D$ variables have been sampled. $N$ denotes the total number of unique timestamps in the series. $\mathbf{X} \in \mathbb{R}^{N \times D}$ constitutes the observation values at the recorded times. $\mathbf{M} \in \{0, 1\}^{N \times D}$ is a binary masking matrix indicating whether a variable has an observation at a specific sampled time. In specific, for each variable $i$ at time $t_n$, if the variable is sampled at time stamp $M^i_{t_n} = 1$, otherwise $M^i_{t_n} = 0$.

### 3.2 OVERALL ARCHITECTURE

The overview of GITaR is illustrated in Figure 2. The primary objective of GITaR is to learn a reconstruction model $\mathcal{R}$ that effectively captures the underlying temporal dynamics in irregular time series data while reconstructing masked portions of the input data. The choice of a reconstruction-based approach is grounded in the fundamental concept of masking and prediction, which aligns closely with the goals of time series regression tasks and facilitates learning complex temporal dynamics. GITaR incorporates two key components: an *irregular-sensitive masking and patching* module, and *an irregular-temporal encoder* module. The first module preserves the original irregular patterns while masking and segments the multivariate time series into synchronized, channel-independent patches. This approach ensures effective learning of local semantics (i.e., irregular patterns) within patches and enhances generalization capability via channel-independent design, reducing processing complexity, and mitigating long-range information loss. The second module aims to: i) learn the local semantic embeddings for each patch through the *irregular patch encoder*, and ii) capture the global temporal correlations among patches via the *global temporal encoder*. The *irregular patch encoder* transforms each irregular patch $p$ into a irregular-aware embedding $Z^i_p$ using continuous time embeddings and irregular time attention mechanisms, leading to a sequence of regularly spaced embeddings. This process ensures local irregularity representation learning. Concurrently, a *global temporal encoder* captures long-range dependencies among $Z^i_p$. These embeddings are then

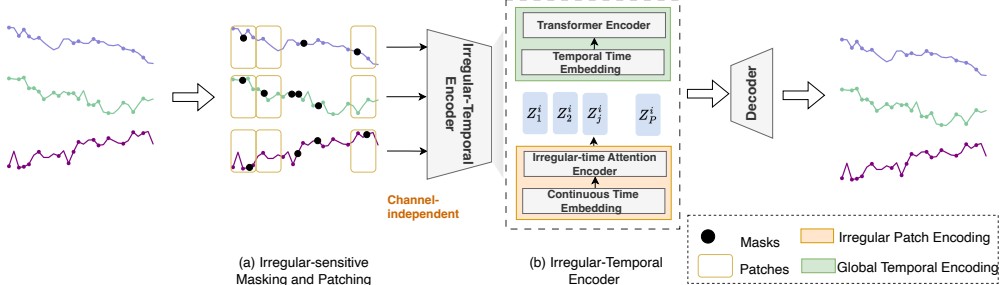

(a) Irregular-sensitive
Masking and Patching

(b) Irregular-Temporal
Encoder

Figure 2: **Workflow of GITaR.** It consists of two modules: (a) Irregular-sensitve masking and patching is designed to segment the multivariate irregular time series while preserving its irregular and sparse characteristics. (b) Irregular-Temporal encoder will map the irregular data into regular representation learning and learn the local semantic embedding for each patch through irregular patch encoding blocks and global temporal dependencies with global temporal encoding block.

decoded and reconstructed to the original space by predicting the masked data. This design enables GITaR to effectively learn the underlying temporal dynamics and semantic information of irregular time series at both local and global levels, facilitating generalized time series regression tasks across various domains and datasets.

### 3.3 IRREGULAR-SENSITIVE MASKING AND PATCHING

To design a more accurate masking and reconstruction SSL framework tailored for irregular time series data, we introduce an irregular-sensitive masking and patching strategy to capture the intrinsic irregularity within the time series, deviating from common random masking techniques (He et al., 2021; Huang et al., 2023) that can distort temporal dependencies in irregular data. Our strategy preserved irregularity patterns and applied channel-independent processing, ensuring application ability to various variables and enabling generalization across different domains.

**Irregular-sensitive Masking.** The irregular-sensitive masking data augmentation aims to maintain the irregular patterns in the original time series. Specifically, recognizing that different regions of the time series may have varying sampling densities, we mask a fixed timespan $q_n$ within each variate rather than a constant number of observations. This approach ensures that in densely sampled regions, more observations are masked, while in sparsely sampled regions, fewer observations are masked. As each variable may have a different sampling frequency and observation gap, we apply the masking strategy independently for each univariate $O^i$.

**Time-sensitive Patching.** In irregular time series data, due to its sparsity, some regions might have more dense observations while others do not. Therefore, simply mapping the whole time series into latent space might lose such local information. We introduce a time-sensitive patching mechanism during the data processing stage to capture local information. Different from the standard time series patching (Nie et al., 2023) that segments regular time series into equal length of subseries-level patches, our approach aims to preserve the synchronization among channels, leading to patches that span fixed time horizons but contain varying numbers of samples. We, therefore, segment each univariate time series into fixed time horizon patches: $O^i = [O_p^i]_{p=1}^{p=P} = (T_p^i, X_p^i, M_p^i)$ for each univariate $i$, where $P$ is the number of resulting patches. This patching strategy allows us to preserve the temporal structure of the irregular time series, enabling efficient processing and local feature extraction and maintaining synchronization across multiple channels.

### 3.4 IRREGULAR-TEMPORAL ENCODING

After the masking and patching stage, each univariate irregular time series has been transformed into a series of patches. However, the irregularity challenge, i.e., the unevenly sampled observations, persists within each chunked patch. To enable the reconstruction model to capture such irregular data, we first design the irregular patch encoding mechanism to learn the embeddings that encode the local irregularity within each chunked patch effectively. Furthermore, to learn the temporal dependencies among these patches, we design the global temporal encoding mechanism to capture the correlation across multiple patches.

**Irregular Patch Encoding.** As each patch is essentially a sub-irregular time series, we employ continuous time embeddings and the irregular-time attention(Shukla and Marlin, 2021b) to encode the patched irregular time series to regular space, aiming to capture the local temporal dynamics. This approach consists of two key components: continuous time embeddings and the irregular-time attention (ITA) mechanism.

Continuous time embeddings function as irregular-aware positional embeddings, which incorporate irregular time points into a fixed vector space by leveraging $H$ embedding functions $\phi_h(t)$, each outputting a representation of size $d_r$. Specifically, Dimension $d$ of embedding $h$ is defined as:

$$\phi_h(t)[d] = \begin{cases} \omega_{0h} \cdot t + \alpha_{0h}, & \text{if } d = 0 \\ \sin(\omega_{dh} \cdot t + \alpha_{dh}), & \text{if } 0 < d < d_r \end{cases}, \quad (1)$$

where $\omega_{dh}$'s and $\alpha_{dh}$'s are learnable parameters. The linear term, when $d = 0$, captures non-periodic patterns that evolve over time, and the periodic terms capture periodicity among time series data.

The subsequent ITA mechanism aims to convert each irregular time series patch $O_p^i$ into irregular-aware embeddings $Z_p^i$ by comparing the irregular time embeddings $\phi_h(t)$ with regular reference points $\mathbf{r}$ using an attention mechanism. Specifically, ITA uses the regular reference time points $\mathbf{r}$ as queries $Q = \phi_h(\mathbf{r})\mathbf{W}_q$, the observed irregular time points $T_p^i$ as keys $K = \phi_h(T_p^i)\mathbf{W}_k$, and the original irregular time series $X_p^i$ as values $V = X_p^i$. The attention mechanism is then employed to obtain the embeddings as follows:

$$Z_p^i = \text{ITA}(\mathbf{r}, O_p^i) = \sum_{h=1}^{H} \left( \text{softmax} \left( \frac{\phi_h(\mathbf{r})\mathbf{W}_q[\phi_h(T_p^i)\mathbf{W}_k]^T}{\sqrt{d_k}} \right) X_p^i \right) \mathbf{W}_l \quad (2)$$

where $\mathbf{W_q}$, $\mathbf{W_k}$ are learnable parameter matrices, $d_k$ is the dimension of the key vectors, and $\mathbf{W}_l$ is a learnable projection vector.

Finally, the irregular patches will be converted to the regular latent space and capture the more fine-grained local information $\mathbf{Z}^i = [Z_p^i]_{p=1}^P$ for each univariate.

**Global Temporal Encoding.** While the encoded patches $\mathbf{Z}^i = [Z_1^i, .. Z_P^i]$ capture local, fine-grained irregularity, learning global temporal correlations between these patches is crucial, especially for time series regression tasks. Therefore, we leverage the transformer multi-head attention mechanism to model the temporal dependency within these patches, where $\mathbf{W} = \{\mathbf{W}_{\hat{q}}, \mathbf{W}_{\hat{k}}, \mathbf{W}_{\hat{v}}\}$, are learnable parameter matrices.

$$\text{MultiHead}(Z^i\mathbf{W}_{\hat{q}}, Z^i\mathbf{W}_{\hat{k}}, Z^i\mathbf{W}_{\hat{v}}) = [\text{head}_1, ..., \text{head}_h]\mathbf{W}, \quad (3)$$

$$\text{head}_h = \text{softmax}(Z^i\mathbf{W}_{\hat{q}}[Z^i\mathbf{W}_{\hat{k}}]^T/\sqrt{d_k}) \quad (4)$$

Then output of MultiHead attention mechanism will be fed into projection layers as in vanilla Transformer encoder to get our final embeddings $E^i$ which capture both local irregular and global temporal dependency information.

## 3.5 RECONSTRUCTION LOSS AND GENERALIZATION

The embeddings $\mathbf{E} = [E^i]_{i=1}^D$ learned from the global temporal encoding will be further decoded by a decoder to reconstruct the original irregular time series $\hat{\mathbf{X}}$. The reconstruction error between the model output $\hat{\mathbf{X}}$ and the target unseen data $\mathbf{X}$ is computed using Mean Squared Error (MSE):

$$L = \frac{1}{D} \sum_{i=1}^{D} (X_i - \hat{X}_i)^2 \cdot M_i \quad (5)$$

This loss function encourages the model to accurately reconstruct the masked portions of the input, thereby learning to capture the underlying patterns and dependencies in the irregular time series data. This pretrained reconstruction model $\mathcal{R}$ will then be fine-tuned for downstream tasks such as forecasting or interpolation on irregular multivariate time series data. To validate its generalization capability, we fine-tune $\mathcal{R}$ across various domains, encompassing different datasets with varying input channels and irregular patterns.

## 4 EXPERIMENTS

**Datasets.** We perform extensive experiments on 4 datasets that span different domains and tasks. **Physionet** (Silva et al., 2012) consists of 37 time series variables extracted from intensive care unit (ICU) records. **MIMIC** (Johnson et al., 2016) consists of electronic health records for more than 60,000 patients with 96 variables. **Human Activity** (Vidulin and Krivec, 2010) has 12 time series variables consisting of irregularly measured 3D positional records. textbfUSHCN (Menne et al.) contains 5 measurements of climate sensing variables. Details are listed in Appendix A.

**Experimental Protocols.** We conduct extensive experiments on multivariate irregular regression tasks including interpolation and forecasting. Specifically, **Interpolation Task** aim to predict missing values within a given time series. **Forecasting Task** im to use the previous data to predict the subsequent future observation values across various variables. Details for are listed in Appendix A.

**Baselines.** We compare GITaR to SOTA time series methods, encompassing both End-to-End supervised training and Self-supervised training approaches. These methods are categorized into 4 groups: (i) E2E for regular time series (E2E-Re), including PatchTST (Nie et al., 2023) and Crossformer (Zhang and Yan, 2023); (ii) SSL for regular time series data (SSL-Re): TS2Vec (Yue et al., 2022); (iii) E2E for irregular time series (E2E-IR), such as mTAND (Shukla and Marlin, 2021b) and t-PatchGNN (Zhang et al., 2024a); and (iv) SSL for irregular time series data (SSL-IR), exemplified by Primenet (Chowdhury et al., 2023). We provide a detailed description in the Appendix A.

**Implementation Details and Metrics.** For both tasks, we select hyper-parameters on the held-out validation set using grid search and then apply the best-trained model to the best set. We randomly divide the dataset into training, validation, and test sets using ratios of 60%, 20%, and 20% same as previous studies (Shukla and Marlin, 2021b; Zhang et al., 2024a). To assess the prediction of continuous targets, we use common regression metrics, such as the mean-average error (MAE), mean-squared error (MSE) and root-mean-squared error (RMSE). We provide a detailed description in the Appendix A.

## 5 RESULTS

### 5.1 IRREGULAR TIME SERIES REGRESSION TASKS PERFORMANCE

**Interpolation Task.** Table 1 presents the performance of GITaR in comparison to baselines on the interpolation task using the PhysioNet dataset. The percentage ranging from 50% to 90% indicates the observed ratio of the time series. The results reveal several key insights. First, our method achieves the lowest RMSE values, with improvements ranging from 20.7% to 14.1% compared to the second best method (PrimNet). This consistent outperformance highlights the robustness and effectiveness of GITaR in interpolating irregular time series data. Additionally, a clear trend emerges when comparing methods designed for irregular data (e.g., mTand, T-patchGNN, Primnet, and GITaR) with those developed for regular time series (PatchTST, Crossformer, TS2Vec) for both E2E and SSL. The former generally outperforms the latter, underscoring the importance of tailoring approaches to the unique challenges posed by irregular data. In particular, E2E regular methods show higher RMSE values and larger performance variances. This suggests that methods assuming continuous and complete sensing often face challenges when confronted with missing values, as their underlying assumptions are violated in the context of irregular time series. The benefits of SSL are apparent in the results. SSL methods (TS2Vec, Primnet, and GITaR) generally perform well, indicating that leveraging large unlabeled datasets to learn general representations can significantly enhance performance on downstream tasks such as interpolation. Among these SSL approaches, GITaR consistently achieves the best performance across all observed ratios. This superior performance demonstrates GITaR's unique ability to capture and leverage complex patterns in irregular time series data, resulting in more accurate interpolation compared to all other methods, including other state-of-the-art self-supervised approaches.

These results highlight the potential of GITaR for real-world applications where irregular and sparse measurements are common, such as in healthcare monitoring or environmental sensing.

**Forecasting Task.** Table 2 presents the forecasting performance of various methods on four datasets: PhysioNet, MIMIC, Human Activity, and USHCN. Our proposed method consistently

Table 1: **Interpolation performance** on the **PhysioNet** for varying percentages of observed data, evaluated using RMSE (X $10^{-2}$), with best and second-best results in **bold** and underlined.

| Training | Model | 50% | 60% | 70% | 80% | 90% |
|---|---|---|---|---|---|---|
| E2E-Re | PatchTST | $6.92 \pm 0.18$ | $6.95 \pm 0.14$ | $8.17 \pm 0.05$ | $8.13 \pm 0.07$ | $8.47 \pm 0.46$ |
| | Crossformer | $6.79 \pm 1.43$ | $6.87 \pm 1.35$ | $7.12 \pm 0.47$ | $7.48 \pm 0.58$ | $7.63 \pm 0.78$ |
| E2E-IR | mTAND | $6.43 \pm 0.11$ | $6.34 \pm 0.45$ | $6.44 \pm 0.43$ | $6.64 \pm 0.66$ | $6.93 \pm 0.66$ |
| | T-patchGNN | $5.32 \pm 0.87$ | $5.21 \pm 0.38$ | $5.04 \pm 0.89$ | $5.29 \pm 0.44$ | $5.68 \pm 0.32$ |
| SSL-Re | TS2Vec | $7.69 \pm 0.12$ | $7.83 \pm 0.22$ | $8.03 \pm 0.15$ | $8.29 \pm 0.12$ | $8.39 \pm 0.43$ |
| SSL-IR | PrimNet | $\underline{4.78 \pm 0.17}$ | $\underline{4.45 \pm 0.02}$ | $\underline{4.97 \pm 0.02}$ | $\underline{5.22 \pm 0.17}$ | $\underline{5.42 \pm 0.15}$ |
| SSL-IR | **GITaR** | $\mathbf{3.79 \pm 0.21}$ | $\mathbf{3.82 \pm 0.25}$ | $\mathbf{4.19 \pm 0.13}$ | $\mathbf{4.58 \pm 0.21}$ | $\mathbf{4.87 \pm 0.11}$ |

achieves the best performance across all datasets and metrics, demonstrating its effectiveness in handling irregular forecasting task. In specific, our method outperforms all baselines across all datasets, with improvements ranging from 4.4% to 8.7% in MSE and 2.3% to 8.7% in MAE compared to the second-best method (typically T-PatchGNN). The results from forecasting tasks align with those from interpolation tasks, demonstrating the effectiveness of SSL approaches over E2E approaches and highlighting the necessity of tailored designs for irregularity modeling.

To further validate the robustness of our approach, we also evaluate the performance under varying observations and forecast horizons. Specifically, we show GITaR continuously outperform SOTA methods when predict different horizons on Physionet dataset in Table 3. The consistent superior performance across all time horizons underscores GITaR's versatility and robustness in handling various forecasting scenarios, making it a reliable choice for a wide range of temporal prediction tasks in irregular time series data.

Table 2: **Forecasting performance** on four real-world irregular datasets evaluated using MSE and MAE, with best and second-best results in **bold** and underlined.

| Training | Model | PhysioNet | | MIMIC | | Human Activity | | USHCN | |
|---|---|---|---|---|---|---|---|---|---|
| | | MSE$\times 10^{-3}$ | MAE$\times 10^{-2}$ | MSE$\times 10^{-2}$ | MAE$\times 10^{-2}$ | MSE$\times 10^{-3}$ | MAE$\times 10^{-2}$ | MSE$\times 10^{-1}$ | MAE$\times 10^{-1}$ |
| E2E-Re | PatchTST | $12.00 \pm 0.23$ | $6.02 \pm 0.14$ | $3.78 \pm 0.03$ | $12.43 \pm 0.10$ | $4.29 \pm 0.14$ | $4.80 \pm 0.09$ | $5.75 \pm 0.01$ | $3.57 \pm 0.02$ |
| | Crossformer | $6.66 \pm 0.11$ | $4.81 \pm 0.11$ | $2.65 \pm 0.10$ | $9.56 \pm 0.29$ | $4.29 \pm 0.20$ | $4.89 \pm 0.17$ | $5.25 \pm 0.04$ | $3.27 \pm 0.09$ |
| E2E-IR | mTAND | $6.23 \pm 0.24$ | $4.51 \pm 0.17$ | $1.85 \pm 0.06$ | $7.73 \pm 0.13$ | $3.22 \pm 0.07$ | $3.81 \pm 0.07$ | $5.33 \pm 0.05$ | $3.26 \pm 0.10$ |
| | T-PatchGNN | $\underline{4.98 \pm 0.21}$ | $\underline{3.84 \pm 0.03}$ | $\underline{1.75 \pm 0.04}$ | $\underline{7.43 \pm 0.09}$ | $2.97 \pm 0.03$ | $\underline{3.45 \pm 0.02}$ | $\underline{5.00 \pm 0.04}$ | $\underline{3.08 \pm 0.04}$ |
| SSL-Re | TS2Vec | $6.04 \pm 0.09$ | $4.48 \pm 0.05$ | $2.71 \pm 0.03$ | $9.53 \pm 0.09$ | $3.08 \pm 0.05$ | $4.13 \pm 0.06$ | $5.35 \pm 0.04$ | $3.27 \pm 0.09$ |
| SSL-IR | PrimeNet | $5.33 \pm 0.62$ | $5.31 \pm 0.68$ | $1.87 \pm 0.05$ | $9.03 \pm 0.29$ | $\underline{2.94 \pm 0.04}$ | $3.56 \pm 0.07$ | $5.08 \pm 0.02$ | $3.22 \pm 0.08$ |
| SSL-IR | **GITaR** | $\mathbf{4.76 \pm 0.18}$ | $\mathbf{3.70 \pm 0.13}$ | $\mathbf{1.72 \pm 0.10}$ | $\mathbf{7.22 \pm 0.09}$ | $\mathbf{2.76 \pm 0.03}$ | $\mathbf{3.15 \pm 0.02}$ | $\mathbf{4.95 \pm 0.05}$ | $\mathbf{3.01 \pm 0.04}$ |

## 5.2 ANALYSIS OF GENERALIZATION CAPABILITIES

To evaluate the generalization ability of our proposed methods, we conducted experiments on cross-domain transfer learning. Specifically, we trained our model on the PhysioNet dataset (37 variables) and tested its performance on three different datasets: MIMIC, Human Activity, and USHCN. This setup allows us to assess how well our model can adapt to new domains with potentially different data characteristics and variable structures. Table 3 presents the results of our cross-domain generalization experiments. It is important to note that some SOTA baselines are not capable of processing different number of variables and, therefore, cannot generlized to other domains, such as the E2E channel-dependent methods including Crossformer, mTAND and T-patchGNN. The performance demonstrates that GITaR consistently outperforms all baselines across all three transfer scenarios. Moreover, SSL-based methods perform better compared with the E2E training manner, which validates the generalization capability of pertaining mechanism More specifically, the superior generalization capability of GITaR can be attributed to several key design choices that set it apart from existing methods. Unlike PrimeNet, which uses channel-dependent processing, GITaR treats each variable independently during the initial stages of processing. This channel-independent

approach allows our model to learn more general representations that are not biased towards specific inter-variable correlation patterns observed in the training data. As a result, GITaR can more easily adapt to new datasets where the relationships between variables may differ. Furthermore, our method incorporates both local irregular pattern learning within patches and global temporal correlation modeling across patches. This dual-scale approach enables GITaR to capture generalizable patterns at multiple levels of abstraction, contributing to its robust performance across domains.

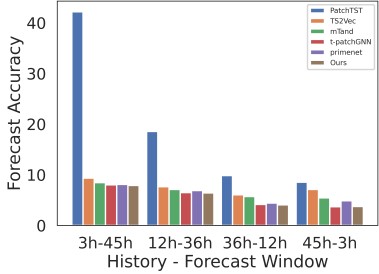

Figure 3: **Forecasting performance** with various prediction horizons.

| **Model** | MIMIC | Human Activity | USHCN |
|---|---|---|---|
| PatchTST | $4.15 \pm 0.13$ | $4.65 \pm 0.26$ | $5.98 \pm 0.07$ |
| TS2Vec | $3.74 \pm 0.21$ | $3.95 \pm 0.13$ | $5.67 \pm 0.08$ |
| PrimeNet | $2.87 \pm 0.34$ | $3.25 \pm 0.16$ | $5.27 \pm 0.05$ |
| **GITaR** | $\mathbf{2.67 \pm 0.17}$ | $\mathbf{2.98 \pm 0.11}$ | $\mathbf{5.14 \pm 0.06}$ |

Table 3: **Comparison of generalization abilities** among different methods, across PhysioNet → MIMIC, or Human Activity or USHCN datasets, indicating training and validation on the former dataset followed by testing on the later dataset.

## 5.3 ABLATION STUDY

We evaluate the performance of GITaR and its several variants on all four datasets we used for regression task. **Complete** represents the model without any ablation; **w/o Ir-mask** removes the irregular-senstive masking strategy while using the random making strategy; **w/o Patch** removes patching module; **w/o GE** (global embedding) removed the temporal and global embedding module and **w/o all** is just the simple masked autoencoder framework.

Table 4 presents the results of the model ablation study. As shown, the removal of any component leads to performance degradation. Notably, the **w/o IR-mask** configuration resulted in a significant performance drop across all datasets, demonstrating the importance of capturing and maintaining the original irregularity when using the masking and reconstruction mechanism. Similarly, the **w/o TS-patch** configuration confirms that patching with fixed horizons or temporal durations preserves local semantics. Additionally, the absence of a transformer for learning global temporal correlations illustrates the necessity of capturing these correlations during representation learning for effective regression forecasting.

Table 4: **Ablation study.**

| Model | PhysioNet | | MIMIC | | Human Activity | | USHCN | |
|---|---|---|---|---|---|---|---|---|
| | MSE$\times 10^{-3}$ | MAE$\times 10^{-2}$ | MSE$\times 10^{-2}$ | MAE$\times 10^{-2}$ | MSE$\times 10^{-3}$ | MAE$\times 10^{-2}$ | MSE$\times 10^{-1}$ | MAE$\times 10^{-1}$ |
| **GITaR (Complete)** | $\mathbf{4.76 \pm 0.18}$ | $\mathbf{3.70 \pm 0.13}$ | $\mathbf{1.72 \pm 0.10}$ | $\mathbf{7.22 \pm 0.09}$ | $\mathbf{2.76 \pm 0.03}$ | $\mathbf{3.15 \pm 0.02}$ | $\mathbf{4.95 \pm 0.05}$ | $\mathbf{3.01 \pm 0.04}$ |
| MAE (**w/o all**) | $7.18 \pm 0.18$ | $4.95 \pm 0.13$ | $3.95 \pm 0.16$ | $12.82 \pm 0.21$ | $4.34 \pm 0.11$ | $4.94 \pm 0.12$ | $5.75 \pm 0.01$ | $3.57 \pm 0.02$ |
| **w/o IR-mask** | $6.81 \pm 0.28$ | $4.76 \pm 0.19$ | $2.95 \pm 0.07$ | $9.85 \pm 0.11$ | $3.34 \pm 0.38$ | $3.94 \pm 0.12$ | $5.33 \pm 0.01$ | $3.36 \pm 0.02$ |
| **w/o TS-patch** | $5.59 \pm 0.67$ | $4.32 \pm 0.46$ | $2.35 \pm 0.56$ | $8.82 \pm 0.43$ | $3.76 \pm 0.59$ | $4.94 \pm 0.32$ | $5.39 \pm 0.20$ | $3.18 \pm 0.09$ |
| **w/o GE** | $5.86 \pm 0.45$ | $4.45 \pm 0.53$ | $2.63 \pm 0.47$ | $9.13 \pm 0.33$ | $3.98 \pm 0.48$ | $5.12 \pm 0.34$ | $5.46 \pm 0.29$ | $3.67 \pm 0.11$ |

## 6 CONCLUSIONS

We have presented GITaR, a novel generalized masking and reconstruction pretraining framework for irregular time series regression. The design of the irregular-sensitive masking and time-patching, the irregular-time encoder, and the global temporal encoding module has shown promise in capturing the underlying dynamics of irregular time series for effective representation learning. More importantly, combined with the channel-independent design, our GITaR has demonstrated superior performance across various domains, effectively leveraging it as a foundational model for multiple tasks or data types involving irregular time series. This work potentially provides a robust and generalised framework for a range of real-world time series applications.

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

# A APPENDIX

## A.1 DATASETS

- **Physionet** (Silva et al., 2012) consists of 37 time series variables extracted from intensive care unit (ICU) records. Each record contains sparse and irregularly spaced measurements from the first 48 hours after admission to ICU. The dataset includes 4000 labeled instances and 4000 unlabeled instances, totaling 8000 instances. We follow the procedures outlined in mTand (Shukla and Marlin, 2021b) for data preprocessing and preparation.

- **MIMIC** (Johnson et al., 2016) consists of electronic health records for more than 60,000 critical care patients. We follow the procedures of Neural Flow (Biloš et al., 2021) and extract 96 time series variables for 48 hours. The datasets include 23418 instances.

- **Human Activity** (Vidulin and Krivec, 2010) has 12 time series variables consisting of irregularly measured 3D positional records from 4 different sensors worn in the waist, check, and ankles. The dataset includes 5 individuals performing various human activities, including walking, sitting, etc.

- **USHCN** (Menne et al.) contains 5 measurements of climate sensing variables over 150 years. We follow the previous work (Biloš et al., 2021) and extract 111,4 station data with a four-year observation period.

- **Semi-synthetic dataset.** we create the semi-synthetic forecasting dataset based on ETTh1 where we introduce irregularity by random dropping data.

## A.2 EXPERIMENTAL PROTOCOLS

**Interpolation Task:** For the interpolation task on the PhysioNet dataset, we aim to predict missing values within a given time series. In specifically, we use 8000 data cases. To evaluate the model's performance under different conditions, we vary the number of observed points used for prediction. Specifically, we test scenarios where 50% to 90% of available data points are used to predict the remaining values, during the test phase.

**Forecasting Task:** For the forecasting task, we aim to use the previous data to predict the subsequent future observation values across various variables. Specifically, during the test time, for Physionet and MIMIC, we use the first 24 hours to predict the next 24 hours. For Human Activity, we use the first 3000 milliseconds to predict the next 1000 milliseconds. For USHCN, we use two years of data to predict the following year.

## A.3 BASELINES:

We employ the following state-of-the-art time series modeling algorithms for comparison.

**PatchTST:** (Nie et al., 2023) enhances the original Transformer architecture by introducing sub-series level patching and channel-independence, effectively capturing cross-time dependencies for time series forecasting.

**Crossformer:** (Zhang and Yan, 2023) modifies the vanilla Transformer architecture by implementing Cross-Time Attention and Cross-Dimension Attention, enabling it to capture both temporal-wise and channel-wise correlations in forecasting tasks.

**mTAND:** (Shukla and Marlin, 2021b) introduces an irregular time series attention mechanism. This method replaces traditional positional embedding with continuous-time embedding, learning correlations between observed data and continuous-time steps to map irregular data onto a fixed, regular representation space.

**T-patchGNN:** (Zhang et al., 2024a) tackles irregular time series forecasting through a multi-step approach. It first employs a transformable patching strategy to segment irregular time series data into uniform temporal resolutions. Subsequently, it utilizes a transformable time-aware convolution network to map the irregular data to a latent space, and finally leverages a graph neural network to learn inter-channel correlations.

**TS2Vec:** (Yue et al., 2022) performs hierarchical contrastive learning over augmented context views, which enables a robust contextual representation for each timestamp. The method is for regular time series classification, regression and anomaly detection task.

**PrimeNet:** (Chowdhury et al., 2023) propose time-sensitive contrastive learning and data reconstruction to learn from data irregularity patterns. The method is for irregular time series classification and interpolation.

### A.4 IMPLEMENTATION DETAILS

Our implementation consists of two primary stages: representation learning and task-specific fine-tuning. During the representation learning stage, we pretrain the model using mask and reconstruction tasks to develop robust representations applicable to various downstream tasks. For each dataset and task, we utilize the entire training set to train the representation model. Subsequently, we fine-tune this model based on the specific task and dataset requirements before evaluating it against the test set to obtain the final representation. To optimize performance, we conduct a comprehensive grid search over key hyperparameters. The batch size is varied among [32, 64, 128], while the learning rate is tested at [0.01, 0.001, 0.0001]. For the model architecture, we explore different configurations: the number of heads (H) in mTAND is varied among [1, 2, 3], and the number of Transformer encoder layers is tested at [2, 4, 6]. To preserve the irregularity within each univariate time series, we implement a masking strategy controlled by the timespan parameter $qn$. This parameter, varied among [2, 3, 4], ensures that the masking ratio for each univariate series remains between 30% and 50%. The choice of patch value, critical for handling data sparsity, is selected from [32, 64, 128] based on the temporal density characteristics of the data. All experiments are run 5 times and the results are reported with mean and standard deviation. All models are implemented using PyTorch, and the experimental evaluations are conducted on NVIDIA A100-SXM-80GB GPUs.

## B SUPPLEMENTARY INFORMATION

### B.1 DATASET STATISTICS

In this section, we provide the updated statistics of our training dataset, as shown in Table 5.

| Dataset | PhysioNet | MIMIC-III | Human Activity | USHCN |
|---|---|---|---|---|
| **Variable Number** | 37 | 96 | 12 | 5 |
| **Missing rate** | 79.6% | 89.1% | 75% | 70.4% |
| **Sampling interval** | 1 hour | 1 hour | 100 seconds | 1 month |

Table 5: Overview of datasets and their characteristics.

## B.2 VISUALIZATION

In this section, we provide updated visualization of attention weights, as shown in Figure 4

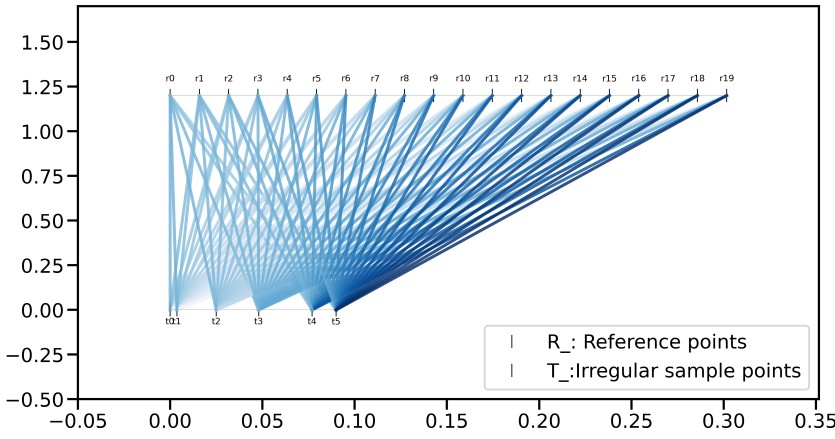

Figure 4: ITA Mapping

## B.3 HYPER-PARAMETER DETAILS

In this section, we provide the details of hyperparameter selections as shown in table 6

| Dataset | Patch Size (P) | Masking Ratio (M) | Final Selection |
|---|---|---|---|
| Physionet | [1, 2, 4, 8, 12] | [0.1, 0.2, 0.3, 0.4, 0.5] | P = 4, M = 0.3 |
| MIMIC-III | [1, 2, 4, 8, 12] | [0.1, 0.2, 0.3, 0.4, 0.5] | P = 4, M = 0.2 |
| Human Activity | [50, 100, 200, 300, 400, 500] | [0.1, 0.2, 0.3, 0.4, 0.5] | P = 200, M = 0.4 |
| USHCN | [1, 2, 4, 8, 12] | [0.1, 0.2, 0.3, 0.4, 0.5] | P = 2, M = 0.2 |

Table 6: Patch size and masking ratio configurations across datasets.

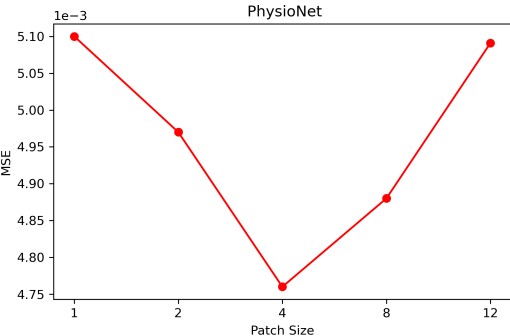

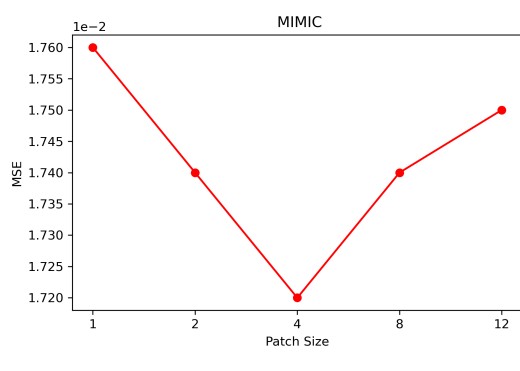

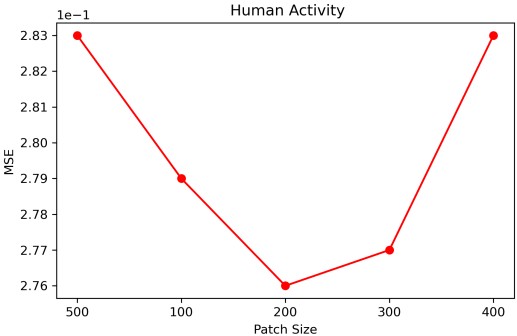

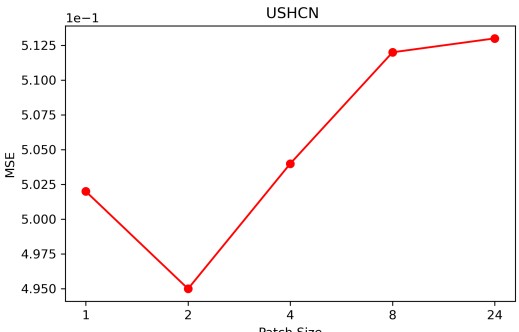

