# OpenReview forum: "GITAR: GENERALIZED IRREGULAR TIME SERIES REGRESSION VIA MASKING AND RECONSTRUCTION PRETRAINING"
_ICLR.cc/2025/Conference — Submitted to ICLR 2025_

### Official Review · Reviewer_hwL2 · 2024-10-28

**Soundness:** 2
**Presentation:** 2
**Contribution:** 2
**Rating:** 3
**Confidence:** 5

**Summary:**

The paper introduces a generalized model for irregular time series regression, leveraging a masking and reconstruction pretraining mechanism. Experimental evaluation is given to show its performance.

**Strengths:**

The proposed model utilizes a masking and reconstruction pretraining mechanism to capture irregular patterns without requiring labeled data, enabling robust and generalized representations.

**Weaknesses:**

1. Although some modules are integrated, the motivation is missing in the paper, such as why this specific masking approach is useful over others for handling irregular time series. The authors are suggested to give more theoretical discussions and an intuitionistic example to strongly motivate the work. Note that in terms of the results reported in Table 3,  the PrimeNet method is able to obtain comparable performance than that of the proposed model in most cases.

2. What is the distinction between the two critical research questions the authors propose? They appear to address the same issue, as a robust self-supervised learning framework inherently implies the ability to adapt to different domains. Could the authors clarify how these questions are fundamentally different?

3. The description for Fig1 lacks clarity, making it difficult to understand the intended meaning. It is unclear whether “v1-v3” represents different variables, tasks, or something else. Additionally, if the figure is meant to illustrate a different domains/transfer learning issue, a more detailed explanation would help readers grasp the specific context and purpose of the comparison.

4. From the method and the experimental part, I cannot find how the proposed model learns and captures the claimed local semantics (i.e., irregular patterns) and global temporal dependencies. Could the authors provide more detail on how these aspects are addressed?

5. In addition to self-supervised learning, there are numerous deep generative models for irregular time series, including GANs and diffusion models [1-3]. The authors have not discussed or compared with these types of methods. Besides, the metrics used in Table 3 are not clearly specified. Also, while the authors claim “across PhysioNet to MIMIC,” in the caption of Table 3, it does not seem to present results specifically for the PhysioNet dataset.


[1] Tashiro, Yusuke, et al. "Csdi: Conditional score-based diffusion models for probabilistic time series imputation." Advances in Neural Information Processing Systems 34 (2021): 24804-24816.

[2] Zhang, Weijia, et al. "Irregular Traffic Time Series Forecasting Based on Asynchronous Spatio-Temporal Graph Convolutional Networks." Proceedings of the 30th ACM SIGKDD Conference on Knowledge Discovery and Data Mining. 2024.

[3] Li, Yan, et al. "Generative time series forecasting with diffusion, denoise, and disentanglement." Advances in Neural Information Processing Systems 35 (2022): 23009-23022.

**Questions:**

As provided in weakness.

---

> ### Author Response · Authors · 2024-11-25
> **Response to Reviewer hwL2 (1/2)**
>
> We appreciate the reviewer's insightful comments. Please see below our responses.
> > W1: Although some modules are integrated, the motivation is missing in the paper, such as why this specific masking approach is useful over others for handling irregular time series. The authors are suggested to give more theoretical discussions and an intuitionistic example to strongly motivate the work. Note that in terms of the results reported in Table 3, the PrimeNet method is able to obtain comparable performance than that of the proposed model in most cases.
>
> A1. The masking aims to preserve the original irregularity patterns which also contain useful information that can be learnt, by keeping dense regions dense and sparse regions sparse. Additionally, dense regions contain richer information, allowing the model to better capture the underlying temporal patterns. In contrast, making dense regions sparse results in the loss of critical information, leading to less effective learning of temporal dynamics.  We will include this in the introduction to motivate the work. Further, we will add the empirical results in our ablation study as in the table below to highlight the effective masking strategy we have over random masking.
>
> | Masking                    | Interpolation RMSE on PhysioNet |
> |----------------------------|----------------------------------|
> | Random Masking             | 5.87                           |
> | Irregular-sensitive Masking (GITaR) | 3.79                   |
>
> To provide an intuitive example, consider a clinical ICU setting where data collection is often limited due to resource constraints, leading to variability in sampling rates and differences in the number of sensors deployed across ICUs. For instance, one ICU may have frequent sampling of vital signs from ten sensors, while another might collect data less frequently from only six sensors. Additionally, ICU data are typically irregularly sampled due to interruptions in monitoring or patient-specific factors. These variations pose significant challenges for existing methods: regular time-series models struggle with irregular sampling, while most irregular time-series methods are not designed to generalize across domains with differing numbers of variables or sampling rates. To address these limitations, we propose a novel self-supervised learning (SSL) framework that simultaneously tackles irregular modeling and generalization, enabling effective transfer of representations between such heterogeneous ICU settings.
>
>
> Our model still shows slightly improvements over PrimeNet in Table 3, and more importantly it shows significant improvements in Table 2.  Additionally, with longer time series, our model demonstrates increased improvement over PrimeNet, as shown in Figure 4.
>
> >W2: What is the distinction between the two critical research questions the authors propose? They appear to address the same issue, as a robust self-supervised learning framework inherently implies the ability to adapt to different domains. Could the authors clarify how these questions are fundamentally different?
>
> A2.We thank the reviewer for pointing out the potential overlap in the two research questions. We will revise them in the manuscript as follows:
> - How can we effectively model latent representations of irregular time series, overcoming the challenges posed by their inherent variability and lack of temporal structure?
>  - How can we improve the generalization of learned representations to facilitate their application to unseen domains with differing characteristics, such as varying sampling patterns or variable sets?
>
> The first research question focuses on modeling the latent representation of irregular time series, addressing a key challenge that most self-supervised learning (SSL) methods face due to the inherent variability and lack of temporal structure in these datasets. This question is centered on designing effective mechanisms, such as time embeddings and masking strategies, to represent irregular time series in a way that captures both local and global temporal dependencies.
>
> The second research question focuses on improving the generalization of these learned representations to enable their application to unseen domains with differing irregular characteristics. This involves ensuring that the representations can adapt to datasets with varying sampling patterns, missing data rates, and variable sets, making them robust across a wide range of real-world scenarios.

---

> ### Author Response · Authors · 2024-11-25
> **Response to Reviewer hwL2 (2/2)**
>
> >W3. The description for Fig1 lacks clarity, making it difficult to understand the intended meaning. It is unclear whether “v1-v3” represents different variables, tasks, or something else. Additionally, if the figure is meant to illustrate a different domains/transfer learning issue, a more detailed explanation would help readers grasp the specific context and purpose of the comparison.
>
> A3. the v1-v3 represent different variables. We will include more details in the figure caption and text, and label the variables in the figure more clearly.
>
> >W4. From the method and the experimental part, I cannot find how the proposed model learns and captures the claimed local semantics (i.e., irregular patterns) and global temporal dependencies. Could the authors provide more detail on how these aspects are addressed?
>
> A4. The model captures **local semantics** by first dividing each time series into patches and mapping each patch to a latent vector consisting of local information. Within each patch, continuous time embeddings are used to preserve the temporal irregularities, which are then processed by a Gated Recurrent Unit (GRU) network. The GRU effectively learns the irregular patterns and dependencies within each patch, outputting a single latent representation for the patch.
>
> For **global temporal dependencies**, we use a transformer-based architecture to integrate the latent representations from all patches, capturing the temporal dynamics among the sequence of patches. We will revise the manuscript to include these details in the methods section and reference the relevant experimental and ablation studies that demonstrate the effectiveness of these modules.
>
> >W5. In addition to self-supervised learning, there are numerous deep generative models for irregular time series, including GANs and diffusion models [1-3]. The authors have not discussed or compared with these types of methods.
> Besides, the metrics used in Table 3 are not clearly specified.
>
> A5. We appreciate the reviewer’s suggestion. We focus on self-supervised learning and E2E supervised compared to baselines that fall under this scheme, as outlined in [1,2]. While GANs and diffusion models also process irregular time series with specific mechanisms, they are based on generative modeling approaches, and fall outside the scope of this work.
>
> We will specify the evaluation metrics used in Table 3 (MSE for the forecasting task) and revise the visualization and explanation to make the table clearer.
>
> [1]Chowdhury, Ranak Roy et al. “PrimeNet: Pre-training for Irregular Multivariate Time Series.” AAAI Conference on Artificial Intelligence (2023).
>
> [2]Zhang, Weijiao et al. “Irregular Multivariate Time Series Forecasting: A Transformable Patching Graph Neural Networks Approach.” International Conference on Machine Learning (2024).
>
> >Also, while the authors claim “across PhysioNet to MIMIC,” in the caption of Table 3, it does not seem to present results specifically for the PhysioNet dataset.
>
> Table 3 focuses on generalization experiments where PhysioNet is used as the pretraining dataset, and the downstream evaluation datasets include MIMIC-III, Human Activity, and USCHN. The results on physionet itself (pretraining and evaluation within the same dataset) is listed in Table 2.
> We will revise the manuscript to clarify this experimental setup and improve the description of the results in both tables to avoid any confusion.

---

> > ### Comment · Reviewer_hwL2 · 2024-11-28
> >
> > I appreciated your response and have carefully read both the rebuttal and the revised paper. However, most of my concerns remain unaddressed. Therefore, I would like to maintain my score. Specially:
> >
> > **For A1 & A5:** From a high-level perspective, I am unable to discern the advantages of the proposed method compared to numerous deep generative models for irregular time series, including GANs and diffusion models. The motivation for this approach is missing in the paper. While the authors responded, ***“We focus on self-supervised learning,”*** and ***“While GANs and diffusion models also process irregular time series with specific mechanisms, they are based on generative modeling approaches, and fall outside the scope of this work,”*** this explanation is not convincing. Why not compare with current SOTA generative models, such as diffusion-based approaches? Additionally, I carefully reviewed the abstract and introduction, the authors discuss recent advances in self-supervised learning (SSL) in the final paragraph but fail to mention related work on generative models for irregular time series earlier in the paper. Unless the authors demonstrate that SSL is the current SOTA solution for irregular time series and outperforms models like diffusion-based models.
> >
> > **Based on the above: For A2**, if the goal is to model latent representations of irregular time series, the comparison should explicitly include latent representations from generative models. If the objective is to improve the generalization of learned representations, the paper should compare and validate against methods like few-shot or zero-shot representation learning. This lack of clarity in the positioning of the work makes the overall motivation and contributions confusing.
> >
> > **For A4:** My question was not about using the framework to explain how local semantics (i.e., irregular patterns) and global temporal dependencies are learned and captured. I was asking how the experiments demonstrate or provide evidence that the model has indeed learned these local semantics and global dependencies.

---

> > > ### Author Response · Authors · 2024-12-01
> > >
> > > Thank you for the additional feedback and here is our response to your comments.
> > >
> > > > Q1. For A1 & A5: From a high-level perspective, I am unable to discern the advantages of the proposed method compared to numerous deep generative models for irregular time series, including GANs and diffusion models. The motivation for this approach is missing in the paper. While the authors responded, “We focus on self-supervised learning,” and “While GANs and diffusion models also process irregular time series with specific mechanisms, they are based on generative modeling approaches, and fall outside the scope of this work,”this explanation is not convincing. Why not compare with current SOTA generative models, such as diffusion-based approaches?
> > >
> > > A1. Thank you for pointing out the related work and motivation of our work.
> > > We appreciate your suggestion to compare our method with generative models, such as diffusion models, which are indeed a promising class of models for time series[1]. Upon reviewing the literature, we found one relevant work [2] that explores diffusion models for irregular time series. While this is an interesting approach, one of our baselines, PrimeNet, already outperforms this method on the interpolation task using the PhysioNet dataset. Additionally, t-PatchGNN, another of our baselines, also employs a diffusion-based model as an encoder to model irregular patterns, as you suggested [3]. That said, we acknowledge the potential of generative models like diffusion-based approaches for irregular time series and agree that there is room to explore their use further. However, we believe that the ability of current diffusion models to effectively handle multivariate time series and generalization still requires further enhancement [1]. Therefore, we chose to focus on SSL for this work, as we believe it directly addresses key challenges in handling irregular time series, particularly in terms of generalization across domains. This, however, is a promising avenue for future work, and we would be happy to explore it further.
> > >
> > > We follow the extensive evaluation framework established by recent works in the field, such as PrimeNet and t-PatchGNN, which also focus on similar datasets and baseline comparisons. We will update the manuscript to include more detailed discussion of these related works in the Related Work and Discussion sections to clarify our position.
> > >
> > > We are also open to any suggestions from the reviewer regarding potential diffusion-based baselines.
> > >
> > > [1]Yang, Yiyuan et al. “A Survey on Diffusion Models for Time Series and Spatio-Temporal Data.”
> > > [2]Shirzad et all “CONDITIONAL DIFFUSION MODELS AS SELFSUPERVISED LEARNING BACKBONE FOR IRREGULAR TIME SERIES” TS4H@ICLR2024
> > > [3]Zhang, Weijia, et al. "Irregular Traffic Time Series Forecasting Based on Asynchronous Spatio-Temporal Graph Convolutional Networks." Proceedings of the 30th ACM SIGKDD Conference on Knowledge Discovery and Data Mining. 2024.
> > >
> > >
> > > >Q2. Additionally, I carefully reviewed the abstract and introduction, the authors discuss recent advances in self-supervised learning (SSL) in the final paragraph but fail to mention related work on generative models for irregular time series earlier in the paper. Unless the authors demonstrate that SSL is the current SOTA solution for irregular time series and outperforms models like diffusion-based models.
> > >
> > > A2. We would like to emphasize that the main contribution of our work is the design of a self-supervised learning (SSL) framework that is capable of generalizing across multiple unseen domains of irregular time series data. This framework is designed to handle a variety of tasks, accommodate different irregular patterns, and be flexible with respect to the number of channels. Our primary focus is on advancing SSL as the core approach for these challenges. As such, a direct performance comparison between SSL and diffusion models falls outside the scope of this study.
> > >
> > > >Q3. Based on the above: For A2, if the goal is to model latent representations of irregular time series, the comparison should explicitly include latent representations from generative models. If the objective is to improve the generalization of learned representations, the paper should compare and validate against methods like few-shot or zero-shot representation learning. This lack of clarity in the positioning of the work makes the overall motivation and contributions confusing.
> > >
> > > A. Our key contribution lies in the design of the SSL model. Accordingly, our evaluation adheres to the standard protocols of pre-training, fine-tuning, and inference. While few-shot and zero-shot representations are not the focus of this study, we appreciate the reviewer’s feedback and would like to note that one of our baselines, PrimeNet, does provide insights in this context, and we believe that our approach shares some similar capabilities in terms of generalization. We will add such discussion in the revised manuscript to make the contribution clearer.

---

> ### Author Response · Authors · 2024-12-01
>
> >Q4. For A4: My question was not about using the framework to explain how local semantics (i.e., irregular patterns) and global temporal dependencies are learned and captured. I was asking how the experiments demonstrate or provide evidence that the model has indeed learned these local semantics and global dependencies.
>
> A4. Local and global information correspond to specific segments of the time series. In our approach, each time series is divided into N consecutive patches, which capture partial information and are referred to as local information, independent of the model structure. These patches are then mapped into latent representations that focus on local details. A subsequent transformer module aggregates these representations into a global representation, which captures long-range temporal dependencies through attention mechanisms. Additionally, our ablation study shows that removing the patches or the global aggregation module leads to performance degradation, underscoring the importance of the information learned. We understand the reviewer’s request for experimental evidence in this regard. However, at present, there are no established theoretical methods to quantify temporal dynamics in the latent representations, nor to directly quantify the relationship between the original time series patches and the latent representations. We would greatly appreciate any suggestions from the reviewer on how to address this and demonstrate these aspects more effectively.

---

### Official Review · Reviewer_sAeC · 2024-11-01

**Soundness:** 3
**Presentation:** 2
**Contribution:** 3
**Rating:** 6
**Confidence:** 4

**Summary:**

In this paper, the authors propose an architecture and training scheme for
irregular time series. They propose to patch and mask not specific amounts of
data points, but times-spans that can include more points in denser and lesser
points in sparser regions. Furthermore, the authors propose an architecture
which first use attention inside the patches (Called Irregular Patch Encoding)
and then an attention mechanism between different patches (Global Temporal
Encoding).

The authors test their approach both on regular regression task and on
out-of-domain forecasting.

**Strengths:**

+ The Architecture and the design decisions seem useful and reasonable.
+ The results are promising, as Gitar outperforms t-patchGNN which outperforms a lot of the ODE Models.
+ Ablation study shows that indeed all components are needed

**Weaknesses:**

- My strongest critique point is the paper writing. The paper on the hand contains a lot of self-advertising sentences which distracts from the "real content". Example:
  - Page 4, Starting at 207: "The first module preserves the original irregular patterns
    while masking and segments the multivariate time series into synchronized, channel-independent
    patches. This approach ensures effective learning of local semantics (i.e., irregular patterns) within
    patches and enhances generalization capability via channel-independent design, reducing process-
    ing complexity, and mitigating long-range information loss"
    -> Why not simply say: "For each channel, we randomly mask a time-span of range q_d and we patch by the time-span lengths instead of    by the amount of points in the patch."
   - On the other hand, a lot of crucial information is missing in the paper:
     - The decoder: How does it look like? Is it an MLP? Just one layer?
     - What does the following mean:
        "This pre-trained reconstruction model R will then be fine-tuned for downstream tasks such as
        forecasting or interpolation on irregular multivariate time series data. To validate its generalization
        capability, we fine-tune R across various domains, encompassing different datasets with varying
        input channels and irregular patterns"
        How exactly does that look like? How does the fine-tuning look like? How do you do for example forecasting? With a forecasting MLP- head as in PatchTST? Or do you just assume that the full forecasting horizon is masked and you apply your normal reconstruction?
      - How do you mask? Randomly select a time-span? How long is the time-span? Does masking means zeroing out? I would formalize and explain such crucial points in detail as instead of just mentioning them shallowly.
    - I was also not able to have a look at the code to answer these questions by myself. I could unfortunately not find a link in the paper and also I could not see any supplementary material.

- In the ablation study, I could not find the answer to the following question: Do you need pre-training at all? How good is your model when being only trained on target forecasting tasks?
- Missing comparison method: Can you compare your method against (https://ojs.aaai.org/index.php/AAAI/article/view/29560)? This paper also outperforms all ODE-based models and may be competitive to your model.

## Final Conclusion
The results look promising and I thus think this paper has indeed potential. However, I think the paper writing has to be modified before publishing, as a lot of crucial details are missing.

**Questions:**

- How does your model perform without any pre-training at all? When being just trained on the forecasting tasks?
- How did you collect your baseline results? Do you re-implement all baselines? Because your results for t-PatchGNN, are different then the results reported in the t-PatchGNN paper itself, have a look the table: (https://github.com/usail-hkust/t-PatchGNN). Regarding the results here, t-Patch GNN would be more competitive to your method.
- How exactly does your fine-tuning approach work?
- How exactly does your decoder look like?

---

> ### Author Response · Authors · 2024-11-25
> **Response to Reviewer sAeC (1/2)**
>
> We appreciate the reviewer's insightful comments. Please see below our responses.
>
> >W1 & Q3 & Q4 My strongest critique point is the paper writing. The paper on the hand contains a lot of self-advertising sentences which distracts from the "real content". Example: …
>
> A1. We will update the revised manuscript to simplify and make the method description more clean and precise.
>
> >The decoder: How does it look like? Is it an MLP? Just one layer?
>
> A1.1 The decoder is a simple MLP with layers designed to reconstruct the original time series at masked positions.  We also tried other decoder structures like symmetric encoder-decoder structures and lightweight decoders, which showed similar performance to MLPs.  We chose MLPs for their simplicity and computational efficiency. The loss is computed only for the masked positions, ensuring a focus on reconstructing missing data effectively.
>
> > What does the following mean: "This pre-trained reconstruction model R will then be fine-tuned for downstream tasks such as ,,," How exactly does that look like? How does the fine-tuning look like? How do you do for example forecasting? With a forecasting MLP- head as in PatchTST? Or do you just assume that the full forecasting horizon is masked and you apply your normal reconstruction?
>
> A1.2 **Fine-tuning procedure:** Then during the fine-tuning for the specific forecasting task, we fine-tune the encoder only on historical observed data while adding a forecasting MLP head. This head predicts the future horizon values. The fine-tuning procedure is simple and modular, designed to adapt to downstream tasks such as forecasting and interpolation.
>
> **Forecasting Task:** The whole process is divided into three stages. During the first pre-training stage, we train the model. In the second fine-tuning stage for forecasting tasks, we divide the time series into past observed time series as input and future values as the forecasting horizons, which serve as the labels. Similar to PatchTST, an MLP head is added for fine-tuning. We use the past observed values as input to optimize the model for forecasting future values using the MSE loss. In the third test phase, we evaluate the model by feeding it the past observations and assessing the forecasted values. We will clarify this in the revised version.
>
> > How do you mask? Randomly select a time-span? How long is the time-span? Does masking means zeroing out? I would formalize and explain such crucial points in detail as instead of just mentioning them shallowly.
>
> A1.3  For the masking strategy, we preserve the original irregular pattern and then keep the relatively dense part stay dense and the sparse area stay sparse, as discussed in section Method.
> We select the time span of 3 data points with a fixed masking ratio within the grid of [0.1, 0.2, 0.3 ,0.4,0.5], setting the masked values to zero. We will include the detailed table for hyper-parameters tuning in the appendix.
> | Dataset         | Patch Size (P)             | Masking Ratio (M)         | Final Selection      |
> |-----------------|----------------------------|---------------------------|----------------------|
> | Physionet       | [1, 2, 4, 8, 12]          | [0.1, 0.2, 0.3, 0.4, 0.5] | P = 4, M = 0.3       |
> | MIMIC-III       | [1, 2, 4, 8, 12]          | [0.1, 0.2, 0.3, 0.4, 0.5] | P = 4, M = 0.2       |
> | Human Activity  | [50, 100, 200, 300, 400, 500] | [0.1, 0.2, 0.3, 0.4, 0.5] | P = 200, M = 0.4     |
> | USHCN           | [1, 2, 4, 8, 12]          | [0.1, 0.2, 0.3, 0.4, 0.5] | P = 2, M = 0.2       |
>
> > I was also not able to have a look at the code to answer these questions by myself. I could unfortunately not find a link in the paper and also I could not see any supplementary material.
>
> A1.4 We will provide the code upon acceptance.

---

> > ### Author Response · Authors · 2024-11-25
> > **Response to Reviewer sAeC (2/2)**
> >
> > > W2 & Q1.   In the ablation study, I could not find the answer to the following question: Do you need pre-training at all? How good is your model when being only trained on target forecasting tasks?
> >
> > A2. We do need pre training as it is the first step of whole training process. In specific, the proposed masked and reconstruction method are applied in the pretraining stage since there are no labels involved and is designed to learn a more general representations.
> >
> > To evaluate our model capability without pretraining, we conducted an ablation study comparing the performance of our model with and without pretraining. In the "without pretraining" setting, the model was trained directly on the target forecasting tasks, using masking as an augmentation method. The results, presented in Table below, demonstrate that while the model performs reasonably well without pretraining,more specifically, it outperforms most of the baselines, the pretraining phase provides a significant boost in performance. We will include these ablation results in the revised manuscript to provide a more comprehensive understanding of the role and benefits of pretraining.
> > | Model          | PhysioNet MSE | MIMIC MSE | Human Activity MSE | USHCN MSE |
> > |----------------|---------------|-----------|---------------------|-----------|
> > | GITaR (ours)   | 4.76          | 1.72      | 2.76                | 4.95      |
> > | w/o pretrain   | 5.45          | 2.43      | 3.45                | 5.14      |
> >
> > >W3. Missing comparison method: Can you compare your method against (https://ojs.aaai.org/index.php/AAAI/article/view/29560)? This paper also outperforms all ODE-based models and may be competitive to your model.
> >
> > A3. Thank you for pointing out this graph-based method; we will include it in the related work section. Upon reviewing GraFITs, we found that our method outperforms it in several aspects. For example, in the PhysioNet prediction task (using the past 24 hours to predict the next 24 hours), our method achieves an error of 4.76 × 10⁻³, compared to GraFITs’ 4.01 × 10⁻¹. More importantly, our method is specifically designed to address the generalization problem for unseen domains with varying channel numbers or sampling rates. In contrast, GraFITs relies on modeling the correlations between different channels during training, making it challenging to generalize effectively to unseen domains.
> >
> >
> > >Q2 : How did you collect your baseline results? Do you re-implement all baselines? Because your results for t-PatchGNN, are different then the results reported in the t-PatchGNN paper itself, have a look the table: (https://github.com/usail-hkust/t-PatchGNN). Regarding the results here, t-Patch GNN would be more competitive to your method.
> >
> > A2. Yes, all baselines were reimplemented using official code where available. We followed the implementation and settings in the t-PatchGNN github repo. However, due to the computing platform different from the original paper, leading to slight discrepancies in results. We have included the GPU settings in the Appendix A  on reproducibility and will also provide the code.

---

> > > ### Comment · Reviewer_sAeC · 2024-11-26
> > > **Respone to Rebuttal**
> > >
> > > Dear Authors,
> > > thank you for the extensive rebuttal and the additionally included experiments.
> > >
> > > Assuming that all the additional experiments and explanations are indeed added to the camera-ready, I am happy to increase my score.

---

### Official Review · Reviewer_oiGX · 2024-11-02

**Soundness:** 2
**Presentation:** 2
**Contribution:** 2
**Rating:** 5
**Confidence:** 4

**Summary:**

The paper proposes a new generalized regression framework for irregular time series data.  The model is trained in the self-supervised mode by relying on a patching and masking mechanism that permits to define and learn the model on a variety of regression tasks. The task sampling schema used for training the model is proposed. After the self-supervised training the model can be used to support a variety of downstream regression tasks.  The model is tested on prediction and interpolation tasks on four datasets showing promising prediction performance against baselines.

**Strengths:**

Originality: A novel attempt to train the models supporting regression inferences on irregularly sampled time series data. Also the idea of using mTAN approach for encoding irregular time series patches to regular representation is interesting and can be applicable to future time series neural network architectures

Significance: Irregularly sampled time series data are important for many practical applications, most prominently healthcare. The objective of developing models that are able to support many downstream regression tasks  is well justified.

Experiments: •	Authors show a promise of the GITaR method against baseline methods in interpolation, forecasting and transfer learning tasks on multiple irregular time series datasets.

**Weaknesses:**

**** Methodology ****

The novelty of the patch schema is unclear and very much resembles recent work on T-PatchGNN.

Weijia Zhang, Chenlong Yin, Hao Liu, Xiaofang Zhou, and Hui Xiong. Irregular multivariate time series forecasting: A transformable patching graph neural networks approach. ICML, 2024 I

It appears that the Global Temporal Encoding $E^i$ is over individual time series and does not capture dependencies across different time series. It is unclear if/how these patterns and dependences are captured in the model.

The description of the decoder architecture is missing in the paper. It is understandable that the model transforms the irregular observations to a regular latent representation. It is unclear how the model then makes the predictions of time series variables at irregular times from this regular latent representation.


**** Experiments *****

Evaluating the performance of models designed for regular time series (Re) against those specifically tailored for irregular time series (IR) on irregular time series regression tasks  doesn’t add much value to the analysis. Perhaps authors intend to just highlight the importance of incorporating time explicitly in the model is important, which makes sense. However, including only 3 IR model baselines in the main results section is limiting. More IR baselines from the families: RNN-based (e.g., GRU-D [1]), Graph-based (e.g., Raindrop [2]), and ODE-based (e.g., ContiFormer [3]) should be considered in the experimental analysis for robust evaluation.

[1] Zhengping Che, Sanjay Purushotham, Kyunghyun Cho, David Sontag, and Yan Liu. Recurrent neural networks for multivariate time series with missing values.

[2] Xiang Zhang, Marko Zeman, Theodoros Tsiligkaridis, and Marinka Zitnik. Graph-guided network for irregularly sampled multivariate time series.

[3] Yuqi Chen, Kan Ren, Yansen Wang, Yuchen Fang, Weiwei Sun, and Dongsheng Li. Contiformer: Continuous-time transformer for irregular time series modeling, 2024.

Interpolation experiments are performed only on Physionet dataset, and do not have MAE metric in the final comparison.

The results section on generalization capabilities is promising but limited to draw meaningful conclusions from. It is unclear why training is performed only on the Physionet dataset and tested on the rest. It would be worthwhile to see if the benefits of this generalization hold up for when the model is pretrained on other datasets or on a combination of them and tested on the remaining set. For example, train on MIMIC and test on the rest; train on MIMIC+Physionet and test on the rest would help to evaluate GITaR’s generalization capabilities.

**** Code *****

The code necessary for reproducing the reported results is not included in the submission.

**Questions:**

- Can you explain the difference of patching mechanism in T-PatchGNN and Time-sensitive Patching proposed in your work?

- The text for section on generalization capabilities can be improved. Can you clarify how exactly the results are computed? Is there a transfer learning step (i.e., training) on the target domain? Is it the forecasting task? If so, what is the forecasting horizon? Is it comparable to the forecasting results (i.e., Table 2)?

 Are there any special or extreme cases where task sampling schema for supporting SSL proposed in the paper may fail? If the masking of observation has uniform timespans, a sparsely sampled time series will have a smaller number of samples masked than a densely sampled time series. Because of these unequal masked observations, the model is biased to learn densely sampled time series better than sparsely sampled one.

---

> ### Author Response · Authors · 2024-11-25
> **Response to Reviewer oiGX (1/3)**
>
> We appreciate the reviewer's insightful comments. Please see below our responses.
>
> > W1. The novelty of the patch schema is unclear and very much resembles recent work on T-PatchGNN.
>
> A1. The patch schema that segments long sequences into fixed time  horizons is the same as T-PatchGNN, but the processing for each patch is different in terms of channel dependent and independent processing. We will cite the original paper and clarify this in the revised Method section.
>
> > W2. It appears that the Global Temporal Encoding E^i is over individual time series and does not capture dependencies across different time series. It is unclear if/how these patterns and dependences are captured in the model.
>
> A2.
> We intentionally designed the model to avoid capturing dependencies across channels in multivariate time series. This decision is rooted in our goal of enhancing the model's generalization ability to datasets with varying numbers of channels. By adopting a channel-independent learning strategy, the model learns representations for each channel (or individual time series) separately, enabling it to adapt to new test domains with different numbers of channels.
>
> In contrast, approaches like T-patchGNN specifically capture channel dependencies to model the relationships among a fixed number of channels (N). While this strategy works well when the number of channels in the training and test datasets is consistent, it hinders generalization to datasets with a different number of channels (e.g., N+2). Such dependency-specific learning fails to adapt because the relationships among channels change with variations in channel count.
>
> >W3. The description of the decoder architecture is missing in the paper. It is understandable that the model transforms the irregular observations to a regular latent representation. It is unclear how the model then makes the predictions of time series variables at irregular times from this regular latent representation.
>
> A3. The decoder consists of simple MLP layers designed to reconstruct the complete regular time series representation from the latent embeddings. While the model reconstructs the full time series, we only calculate the loss for the observed points during training and evaluate performance on the observed points during testing. This approach ensures alignment with the irregular nature of the original time series while leveraging the regularized latent representation for reconstruction.
>
> The choice of MLP as the decoder architecture is supported by our experiments, where we compared it with other architectures, including a symmetric encoder-decoder structure and a lightweight encoder-decoder design. These alternatives showed similar performance to the MLP-based decoder, so we selected MLP for its simplicity and computational efficiency. We will add a detailed description of the decoder architecture and our rationale for choosing MLP in the revised manuscript.
>
> >W4.  Evaluating the performance of models designed for regular time series (Re) against those specifically tailored for irregular time series (IR) on irregular time series regression tasks doesn’t add much value to the analysis. Perhaps authors intend to just highlight the importance of incorporating time explicitly in the model is important, which makes sense. However, including only 3 IR model baselines in the main results section is limiting. More IR baselines from the families: RNN-based (e.g., GRU-D [1]), Graph-based (e.g., Raindrop [2]), and ODE-based (e.g., ContiFormer [3]) should be considered in the experimental analysis for robust evaluation.
>
> A4. We appreciate the reviewer’s suggestion to include additional IR model baselines, such as GRU-D, Raindrop, and ODE-based methods like ContiFormer. However, as demonstrated in prior works [1,2], GRU-D and ODE-based methods have been significantly outperformed by more recent approaches. Similarly, Raindrop has been shown to be less competitive compared to T-patchGNN in related studies.
>
> To provide a robust evaluation of our method, we focused on comparing it with the most competitive baselines, as shown in Table 1 & 2. We will include these references and a brief justification for selecting our baselines in the revised manuscript.
>
> [1]Shukla, Satya Narayan and Benjamin M Marlin. “Multi-Time Attention Networks for Irregularly Sampled Time Series.” ArXiv abs/2101.10318 (2020): n. Pag.
>
> [2]Chowdhury, Ranak Roy et al. “PrimeNet: Pre-training for Irregular Multivariate Time Series.” AAAI Conference on Artificial Intelligence (2023).

---

> > ### Comment · Reviewer_oiGX · 2024-11-29
> > **Follow-up comments**
> >
> > Thank you for your responses and clarifications. Unfortunately I do not think all my questions and comments were answered to my full satisfaction Here are the issues:
> >
> > A4. We appreciate the reviewer’s suggestion to include additional IR model baselines, such as GRU-D, Raindrop, and ODE-based methods like ContiFormer. However, as demonstrated in prior works [1,2], GRU-D and ODE-based methods have been significantly outperformed by more recent approaches. Similarly, Raindrop has been shown to be less competitive compared to T-patchGNN in related studies.
> >
> > Comment:  I still believe it is important to provide a rich set of baselines making the claims in one's work. The argument in the response is based on indirect evidence and statements such as method A was shown to outperform method B in some paper. What is forgotten is that such claims are made on a limited number of datasets and these data sets may vary from work to work. For example PrimeNet paper that is cited in your response uses experiment on MIMIC dataset with 12 variables. Does the result hold on MIMIC versions with 97 or more variables.
> >
> > A2. We intentionally designed the model to avoid capturing dependencies across channels in multivariate time series. This decision is rooted in our goal of enhancing the model's generalization ability to datasets with varying numbers of channels. By adopting a channel-independent learning strategy, the model learns representations for each channel (or individual time series) separately, enabling it to adapt to new test domains with different numbers of channels.
> >
> > Thanks for confirming the design of your method and one of its key assumptions. What is intriguing is that all forecasting results in Table 1 and Table 2 on your model  that does not model multi-variate dependences outperforms dedicated forecasting models that capture or at least try to capture these dependences. What is the reason for that?  Just the training sample-size? Assuming the training size is large enough would you expect the model to become inferior?

---

> > > ### Author Response · Authors · 2024-12-02
> > > **Response to Reviewer oiGX's follow-up comments**
> > >
> > > >*A4. We appreciate the reviewer’s suggestion to include additional IR model baselines, such as GRU-D, Raindrop, and ODE-based methods like ContiFormer. However, as demonstrated in prior works [1,2], GRU-D and ODE-based methods have been significantly outperformed by more recent approaches. Similarly, Raindrop has been shown to be less competitive compared to T-patchGNN in related studies.*
> > > **Comment:** I still believe it is important to provide a rich set of baselines making the claims in one's work. The argument in the response is based on indirect evidence and statements such as method A was shown to outperform method B in some paper. What is forgotten is that such claims are made on a limited number of datasets and these data sets may vary from work to work. For example PrimeNet paper that is cited in your response uses experiment on MIMIC dataset with 12 variables. Does the result hold on MIMIC versions with 97 or more variables.
> > >
> > > **Answer:**
> > >
> > > Thank you for your suggestion. We agree that aligning experimental protocols and reproducing baselines under consistent settings is crucial for fair comparison. Regarding your earlier comment, GRU-D and ODE-based models are primarily trained in a supervised end-to-end (E2E) manner, which limits their adaptability in generalization experiments. For this reason, we did not include them in our study. Additionally, for forecasting tasks, we followed the experimental setup similar to t-PatchGNN, which demonstrated superior performance compared to these baselines when using 96 variables in the experiment. We will provide a comprehensive baseline comparison and clarify these details in the Experiments section of the revised manuscript.
> > >
> > > >*A2. We intentionally designed the model to avoid capturing dependencies across channels in multivariate time series. This decision is rooted in our goal of enhancing the model's generalization ability to datasets with varying numbers of channels. By adopting a channel-independent learning strategy, the model learns representations for each channel (or individual time series) separately, enabling it to adapt to new test domains with different numbers of channels.*
> > > **Comment:** Thanks for confirming the design of your method and one of its key assumptions. What is intriguing is that all forecasting results in Table 1 and Table 2 on your model that does not model multi-variate dependences outperforms dedicated forecasting models that capture or at least try to capture these dependences. What is the reason for that? Just the training sample-size? Assuming the training size is large enough would you expect the model to become inferior?
> > >
> > > **Answer:**
> > >
> > > Experimentally, we combined the Physionet and MIMIC-III datasets as training data to pre-train the model. This approach increased the dataset size, and our model still outperformed baseline models that learn channel dependencies, achieving even higher relative improvements of 26% and over CrossFormer, the method that captures the channel-dependence, in Physionet datasets. Besides, this highlights that not learning channel dependencies can be advantageous for generalization.
> > >
> > > Conceptually, we identified two reasons for this:
> > >
> > > * In multivariate irregular time series, different channels or variables often have distinct sampling rates and missing patterns, resulting in data missing randomly across different variables or at various time steps. Existing models designed to capture channel dependencies in regular time series, such as Crossformer, lack effective mechanisms to handle such dynamic missingness across channels. Consequently, they are less effective at capturing the multi-channel patterns inherent in irregular time series. By learning each variable independently, our model avoids these challenges, resulting in superior performance.
> > >
> > > *More importantly, our primary goal is to generalize the pre-trained model to various unseen datasets with differing irregularities and channels. If we perfectly capture channel dependencies during the pre-training stage for 12-lead ECG data, it becomes challenging to generalize to 2-channel PPG data since the data types, number of channels, and channel dependencies differ significantly. To ensure that the pre-trained model generalizes across different data types, irregular patterns, and numbers of channels, learning channel independence is a better approach.
> > >
> > > We have highlighted this in section Results  and will provide further clarification in the revised version.

---

> ### Author Response · Authors · 2024-11-25
> **Response to Reviewer oiGX (2/3)**
>
> >W5. Interpolation experiments are performed only on Physionet dataset, and do not have MAE metric in the final comparison.
>
> A5. We followed the experimental setup of the baseline studies and conducted interpolation experiments primarily on the PhysioNet dataset to ensure a consistent and fair comparison. While the current results focus on the metrics used in prior work, we acknowledge the importance of including Mean Absolute Error (MAE) as an additional evaluation metric. We will incorporate the MAE metric into the revised manuscript for a more comprehensive comparison.
>
>
> >W6.  The results section on generalization capabilities is promising but limited to draw meaningful conclusions from. It is unclear why training is performed only on the Physionet dataset and tested on the rest. It would be worthwhile to see if the benefits of this generalization hold up for when the model is pretrained on other datasets or on a combination of them and tested on the remaining set. For example, train on MIMIC and test on the rest; train on MIMIC+Physionet and test on the rest would help to evaluate GITaR’s generalization capabilities.
>
> A6. PhysioNet dataset has a relatively high number of variables and a low missing ratio compared to the other datasets. This makes it a strong candidate for pretraining. To address the reviewer’s concerns, we conducted additional experiments as suggested. Specifically, we trained the model on MIMIC-III and evaluated it on the other three datasets, as well as pretraining on a combined dataset of MIMIC-III and PhysioNet and testing on the remaining datasets. The results are summarized in the tables below.
>
> | Model (Train on MIMIC)         | PhysioNet (MSE x 10-3) | Human Activity (MSE x 10-2) | USHCN (MSE x 10-1) |
> |---------------|--------------------------|------------------------------|----------------------|
> | PatchTST      | 14                       | 6.57                         | 6.53                 |
> | TS2Vec        | 8.12                     | 5.37                         | 6.42                 |
> | PrimeNet      | 6.87                     | 4.24                         | 6.14                 |
> | GITaR (ours)  | 5.89                     | 3.48                         | 5.89                 |
>
> | Model (Train on MIMIC + PhysioNet)      | PhysioNet (MSE x 10-3) | Human Activity (MSE x 10-2) | USHCN (MSE x 10-1) |
> |---------------|--------------------------|------------------------------|----------------------|
> | PatchTST      | 12                       | 4.57                         | 6.33                 |
> | TS2Vec        | 7.63                     | 4.37                         | 6.12                 |
> | PrimeNet      | 5.87                     | 3.42                         | 5.89                 |
> | GITaR (ours)  | 4.97                     | 2.59                         | 5.26                 |
>
> >W7.The code necessary for reproducing the reported results is not included in the submission.
>
> Answer:
> We will provide the code in the camera-ready version.
>
>
>
> In summary, our approach consistently outperforms other baselines across all settings, demonstrating strong generalization capabilities. While pretraining on MIMIC-III alone yielded competitive results, the dataset's high missing ratio presented unique challenges, limiting the extent of generalization compared to other configurations.In contrast, pretraining on MIMIC-III combined with PhysioNet showed improved performance, as the additional data from PhysioNet helped train a more generalized and robust representation.
>
> These results highlight the importance of both the quality and quantity of pretraining data in achieving strong generalization across diverse datasets. We will include these additional experiments and results in the revised manuscript to provide a more comprehensive evaluation of our method’s generalization capabilities.

---

> > ### Author Response · Authors · 2024-11-25
> > **Response to Reviewer oiGX (3/3)**
> >
> > >Q2. The text for section on generalization capabilities can be improved. Can you clarify how exactly the results are computed? Is there a transfer learning step (i.e., training) on the target domain? Is it the forecasting task? If so, what is the forecasting horizon? Is it comparable to the forecasting results (i.e., Table 2)?
> >
> > A2. We will clarify the experiment setup in the revised pdf. Similar to the interpolation and forecasting task,in the generalization task, we first pretrain the model on the source domain (e.g., the physionet dataset) and then finetune the encoder to the target domain (e.g., MIMIC or Activity dataset listed in the table). It is till the forecasting task, and the forecasting horizon is the same as table 2, where we listed the details in the Appendix A.
> >
> > The performance on this generalizable setting is not as good pretrain on the same dataset and then finetune for the different dataset, due to the significantly variability across different datasets, but this is still the first study that highlights the generalization ability of our training mechanism to unseen domains or datasets with irregular time series.
> >
> >
> > >Q3. Are there any special or extreme cases where task sampling schema for supporting SSL proposed in the paper may fail? If the masking of observation has uniform timespans, a sparsely sampled time series will have a smaller number of samples masked than a densely sampled time series. Because of these unequal masked observations, the model is biased to learn densely sampled time series better than sparsely sampled one.
> >
> > A3. Each dataset in Table 3 of our paper has a high and varied rate of missing data, with statistics listed in Table stas in Appendix B. For Human Activity with more dense regions and MIMIC-III with more sparse regions, similar improvements of 34% and 28% are observed. This indicates that our method is not biased towards learning specifically from densely sampled time series.
> >
> > To further validate it, we compared our masking strategy with random masking which leads to more masking in dense regions. The results in the table below demonstrated the superiority of our method. This is likely because dense regions contain richer information, allowing the model to better capture the underlying temporal patterns. In contrast, making dense regions sparse results in the loss of critical information, leading to less effective learning of temporal dynamics.
> > | Masking                    | Interpolation RMSE on PhysioNet |
> > |----------------------------|----------------------------------|
> > | Random Masking             | 5.87                           |
> > | Irregular-sensitive Masking (GITaR) | 3.79                   |
> >
> > We will include these experimental results and the rationale for our masking strategy in the revised manuscript to provide further clarity and evidence supporting our approach.

---

### Official Review · Reviewer_X6xD · 2024-11-03

**Soundness:** 3
**Presentation:** 3
**Contribution:** 3
**Rating:** 5
**Confidence:** 3

**Summary:**

In order to handle irregular multivariate time series data, the paper presents GITAR, a self-supervised learning (SSL) framework that combines irregular-temporal encoder, time-sensitive patching, and irregular-sensitive masking. The suggested model aims to address drawbacks of current approaches, including limited adaptability to new domains and domain-specific inter-channel dependencies. According to tests conducted on a number of real-world datasets, GITAR outperforms other state-of-the-art (SOTA) techniques in irregular time series regression tasks.

**Strengths:**

- The paper addresses a relevant and challenging issue in irregular time series analysis, which is applicable across a variety of fields.

- The framework is technically well-executed with clear mathematical formulations. The irregular-time attention mechanism (ITA) and continuous time embeddings are rigorously defined, providing robustness for learning temporal dynamics.

- The experiments cover multiple datasets, which helps evaluate the framework across different irregular time series contexts.

**Weaknesses:**

- The theoretical foundations of the proposed methods are weakly articulated. The notation and assumptions behind the continuous embeddings and ITA mechanism are not thoroughly explained. Key concepts, such as the initialization and constraints of embedding parameters, are underexplored.

- The model’s generalization claim would be better supported by comparing it to a broader array of baselines that specifically address irregular time series and variable sampling rates.

- The model’s main innovations appear to be modest adaptations of existing techniques (e.g., masked autoencoders). Without stronger theoretical or empirical evidence, GITAR’s contribution to the field of irregular time series analysis seems limited.

**Questions:**

- In case of continuous time embeddings, why are specific sinusoidal terms chosen over other possible basis functions? What benefits do $\omega$ and $\alpha$ provide in capturing irregular periodicities? Is there theoretical support indicating they are optimal for this purpose? How do these embeddings behave under highly irregular sampling intervals?

- What theoretical or empirical basis supports the choice of attention as the primary mechanism for learning temporal dependencies in irregular time series? Would other mechanisms (e.g., kernel-based methods,  spectral embeddings, or Neural ODE/SDE) potentially provide similar or improved performance?

- Is there any analysis or empirical validation demonstrating that these attention weights maintain meaningful interpretations across different irregularity patterns?

- The choice of patch size and masking ratio is critical for the performance of GITAR. How does the model’s accuracy change as these hyperparameters vary? Additionally, how are these hyperparameters selected for datasets with different irregularity patterns?

- Given the high complexity of long time series sequences, how does GITAR address computational challenges associated with global attention mechanisms?

---

> ### Author Response · Authors · 2024-11-25
> **Response to Reviewer X6xD (1/4)**
>
> We appreciate the reviewer's insightful comments. Please see below our responses.
>
> >W1:  The theoretical foundations of the proposed methods are weakly articulated. The notation and assumptions behind the continuous embeddings and ITA mechanism are not thoroughly explained. Key concepts, such as the initialization and constraints of embedding parameters, are underexplored.
>
> A1. The continuous time embeddings and ITA mechanisms are designed to replace traditional positional embeddings to better handle irregular time series, and to map the original irregular time series into regularly sampled latent vectors at fixed intervals. We will include the high-level justification about the assumptions in section Method.
> Initializations are similar to conventional positional embedding in vanilla transformers and there’s no constraints of these parameters, similar to [1]. We will highlight these connections in the revised Method section.
>
> [1] Shukla, Satya Narayan and Benjamin M Marlin. “Multi-Time Attention Networks for Irregularly Sampled Time Series.” ArXiv abs/2101.10318 (2020): n. pag.
>
> >W2. The model’s generalization claim would be better supported by comparing it to a broader array of baselines that specifically address irregular time series and variable sampling rates.
>
> A2. None of the baselines addresses the generalization across various sampling rates and irregular patterns, so we included the SOTA baselines that address both regular and irregular time series, under supervised and unsupervised setting for a comprehensive comparison, as shown in Table 3. These experiments address different irregularity patterns and sampling rates across datasets, as shown in the Table below as well as in **Appendix B**. We can include it in the revised manuscript for clarity.
>
> | Dataset         | PhysioNet | MIMIC-III | Human Activity | USHCN      |
> |-----------------|-----------|-----------|----------------|------------|
> | Variable Number | 37        | 96        | 12             | 5          |
> | Missing rate    | 79.6%     | 89.1%     | 75%            | 70.4%      |
> | Sampling interval | 1 hour    | 1 hour    | 100 seconds    | 1 month    |
>
> >W3. The model’s main innovations appear to be modest adaptations of existing techniques (e.g., masked autoencoders). Without stronger theoretical or empirical evidence, GITAR’s contribution to the field of irregular time series analysis seems limited.
>
> A3. Our key novelty lies in designing a SSL framework that can generalize across various irregular patterns, ratios, and number of signal channels. Existing methods using masked autoencoders have been limited to a fixed number of channels, and have been fine-tuned and evaluated using the same datasets, failing to generalize to different unseen domains. In contrast, our method incorporates channel-independent design and irregular-sensitive masking encoder to efficiently learn the underlying patterns of irregular time series regardless of the characteristics of irregular training data. We demonstrate its generalizability performance at table 3.

---

> ### Author Response · Authors · 2024-11-25
> **Response to Reviewer X6xD (2/4)**
>
> >Q1.1 In case of continuous time embeddings, why are specific sinusoidal terms chosen over other possible basis functions?
>
> A1.1 Sinusoidal terms preserve periodic nature, smoothness, and continuity, functioning similarly to other potential basis functions like cosine functions for position inference. In time embeddings, the primary goal is to map the time index to a vector space while preserving the sequential order, which sinusoidal terms accomplish effectively. Furthermore, when the time indices are discrete, these embeddings can subsume the traditional positional embeddings used in transformers, evident in [1].
>
> [1] Shukla, Satya Narayan and Benjamin M Marlin. “Multi-Time Attention Networks for Irregularly Sampled Time Series.” ArXiv abs/2101.10318 (2020): n. pag.
>
>
> >Q1.2 What benefits do $\omega$ and $\alpha$ provide in capturing irregular periodicities?
>
> A1.2 $\omega$ and $\alpha$ control the frequency and phase in the time embeddings (Eq. xx), and this is simply a mapping from the original time index to a vector space, not a learning of irregular periodicities.
>
> The subsequent step, the ITA mechanism, learns the irregular periodicities. Specifically, given a regular sequence of reference points $r$ such as 1 to 128, and the time embeddings of the original time index such as $\phi = [\phi_1, \phi_2, \ldots, \phi_{10}]$, the attention score will be calculated as:
>
> $$
> \text{Score} = \text{Attention}(r, \phi)
> $$
> This attention score clearly reflects the similarity in time; the nearer the time points, the higher the score. While some time steps in $\phi$ are missing due to the irregularity, the attention score will be manually set to 0 at those missing positions. Finally, the ITA embeddings are calculated by multiplying the attention scores with the original time series values:
>
> $$
> \text{ITA} = \text{Score} \times \text{Values}
> $$
> The ITA embeddings therefore capture information only from the observed points, while automatically taking into account the salient information in the time series. We will break down the description in Section Method, to clarify this.
>
> >Q1.3 Is there theoretical support indicating they are optimal for this purpose?
> A1.3 We have shown the empirical results in Table 1&2, for example, patchTST and Crossformer utilize the conventional position embeddings as in the vanilla transformers, and show inferior performance compared to other approaches.
>
> >Q1.4 How do these embeddings behave under highly irregular sampling intervals?
>
> A1.4 The time embeddings are designed to be robust to highly irregular sampling intervals, as shown in datasets such as MIMIC-III (see the Statistics Table), where the missing ratio and irregular sampling rates are high and our method still demonstrates promising result as shown in Table 1&2. This robustness stems from both the theoretical construction of the embeddings and their integration with the attention mechanism.
>
> High irregularity in sampling intervals does not distort the embeddings because the embedding function only depends on the observed time points themselves, not on their spacing relative to other points. In detail, The continuous embeddings map each observed time point independently to a high-dimensional vector space, as shown in **Figure.ITA_Mapping in Appendix B**. Additionally, the irregular sampling intervals are explicitly addressed by the attention mechanism. The attention weights are computed only for observed points, ensuring that missing or sparse time points do not introduce noise into the representation.

---

> > ### Author Response · Authors · 2024-11-25
> > **Response to Reviewer X6xD (3/4)**
> >
> > >Q2.1 What theoretical or empirical basis supports the choice of attention as the primary mechanism for learning temporal dependencies in irregular time series?
> >
> > A2.1 The foundational idea of the ITA module is to map irregular time series into a vector space with regular time steps while preserving the original temporal information.In detail, the attention mechanism captures the temporal similarity between irregularly sampled time points t and a set of regular reference points r, ensuring the relationship between observed data and its temporal context is preserved. For example, given irregular time indexes $t=[1,5,6,9,10]$ and regular reference points $r=[1,2,3,4,5,6,7,8,9,10]$, the attention scores quantify the similarity between these two sets. More specifically, the embedding procedure happens as follows: Each irregular time point is first transformed using a sinusoidal mapping sin⁡(t) to encode periodic temporal features, and then further embedded through a learnable transformation $f(sin⁡(t))$, which retains the temporal information in a high-dimensional space. The similarity is computed between these transformed embeddings $f(sin⁡(t))$ and the embeddings of the reference points, where closer temporal proximity between $t$ and $r$ results in higher attention weights. As shown in **Figure. ITA_Mapping in Appendix B**, we show the attention weights between irregular timestamps and mapped regular space. The darker lines represent higher attention weights, showing the alignment between irregular points and their closest reference points.
> >
> > These regular latent vectors facilitate the processing of subsequent modules, such as transformers or other neural architectures optimized for regular time series, allowing them to operate effectively.
> >
> > >Q 2.2 Would other mechanisms (e.g., kernel-based methods, spectral embeddings, or Neural ODE/SDE) potentially provide similar or improved performance?
> >
> > A 2.2 Compared to kernel-based methods[1] which use fixed functional forms to learn,the attention mechanism is more flexible to capture the similarity and dependencies between original observed timepoints and mapped time points[2]. For spectral embeddings which need to carefully design similarity matrix for each dataset, which may not generalize well across different datasets and irregular sampling rate. Comapring with Neural ODE/SDE, the attention mechanism have demonstrated superior performance over many ode-based approaches as shown in [2].
> >
> > [1] Li, Steven Cheng-Xian and Benjamin M Marlin. “Classification of Sparse and Irregularly Sampled Time Series with Mixtures of Expected Gaussian Kernels and Random Features.” Conference on Uncertainty in Artificial Intelligence (2015).
> >
> > [2] Shukla, Satya Narayan and Benjamin M Marlin. “Multi-Time Attention Networks for Irregularly Sampled Time Series.” ArXiv abs/2101.10318 (2020): n. pag.
> >
> > > Q3. Is there any analysis or empirical validation demonstrating that these attention weights maintain meaningful interpretations across different irregularity patterns?
> >
> > A3. As per the response to  Q2, the attention weights capture the temporal similarity between irregular sampled time points and a set of regular reference points. More specifically, This design ensures robustness to irregular patterns because the attention scores are computed only for observed time points, while missing observations are masked with a binary matrix, contributing zero to the attention computation. As a result, different irregular sampling patterns, such as varying densities or large gaps, do not distort the computed attention scores. Instead, the mechanism focuses on the available data and aligns it with the temporal context provided by the regular reference points.
> >
> > Empirically, the attention mechanism has demonstrated its robustness and effectiveness across datasets with diverse irregularity patterns, such as PhysioNet and MIMIC-III. This suggests that the attention scores remain interpretable and meaningful regardless of the irregular sampling in the input data.

---

> > > ### Author Response · Authors · 2024-11-25
> > > **Response to Reviewer X6xD (4/4)**
> > >
> > > >Q4. The choice of patch size and masking ratio is critical for the performance of GITAR. How does the model’s accuracy change as these hyperparameters vary? Additionally, how are these hyperparameters selected for datasets with different irregularity patterns?
> > >
> > > A4. We have conducted grid search for hyperparameters including the patch size and masking ratio for each dataset, as shown in the table below. We will add the detailed hyper-parameter tuning table in ** Appendix B**.
> > >
> > > | Dataset         | Patch Size (P)             | Masking Ratio (M)         | Final Selection      |
> > > |-----------------|----------------------------|---------------------------|----------------------|
> > > | Physionet       | [1, 2, 4, 8, 12]          | [0.1, 0.2, 0.3, 0.4, 0.5] | P = 4, M = 0.3       |
> > > | MIMIC-III       | [1, 2, 4, 8, 12]          | [0.1, 0.2, 0.3, 0.4, 0.5] | P = 4, M = 0.2       |
> > > | Human Activity  | [50, 100, 200, 300, 400, 500] | [0.1, 0.2, 0.3, 0.4, 0.5] | P = 200, M = 0.4     |
> > > | USHCN           | [1, 2, 4, 8, 12]          | [0.1, 0.2, 0.3, 0.4, 0.5] | P = 2, M = 0.2       |
> > >
> > > For the masking ratio, given the already high missing ratio in the irregular time series, we optimized it within the range of 0.1 to 0.5 instead of higher ratio above 0.5 to prevent further loss of information. For the patch size, due to the higher sampling rate in the HAR dataset and the less fine-grained labels, we optimized the patch size within a higher range.
> > > For all four datasets, we observed a decrease in MSE in model performance until the optimal parameters were reached, followed by an increasing trend in MSE. We have updated the ablation study figures in **Appendix B** of the revised version
> > >
> > > >Q5. Given the high complexity of long time series sequences, how does GITAR address computational challenges associated with global attention mechanisms?
> > >
> > > A5. While computational challenges in long time series sequences are indeed an important topic, this is beyond the scope of our work. This paper addresses the challenge of generalization in irregular time series data which is a very understudied topic per se.

---

> > > > ### Comment · Reviewer_X6xD · 2024-11-29
> > > >
> > > > I appreciate the authors' thorough response to my concerns. Thank you for the clarifications in the revision and answers. I am revising my evaluation with a higher score. However, I still have some comments to improve.
> > > >
> > > > - I understand the choice of model configuration intuitively, but lack of theoretical analysis remains a significant gap. Also, current sensitivity analysis is quite limited, because there is only trend in accordance to the patch size. It is understandable to select hyperparameter, but still questionable to its robustness. Is it related to the data characteristic? How can we select the configuration based on theoretical analysis (or other perspective)?
> > > >
> > > > - Did you bound the range of $\omega$ and $\alpha$ in the training process? It seems that these parameter can be fluctuated depending on each samples (or even each variables). How can we guarantee the convergence of these learnable parameters and corresponding ITA values?
> > > >
> > > > - For the comparison with, mTAND in Table 1, it seems the results are different from the original paper [1]. Could you elaborate the reason? For instance, I think the error should be decreased when the observation ratio is increased, but the results in the manuscript are opposite. Furthermore, recent work on Neural ODE/SDE shows better performance in the same task [2].
> > > > [1] Shukla, S. N., & Marlin, B. M. (2021). Multi-time attention networks for irregularly sampled time series. arXiv preprint arXiv:2101.10318.
> > > > [2] Oh, Y., Lim, D., & Kim, S. (2024). Stable Neural Stochastic Differential Equations in Analyzing Irregular Time Series Data. arXiv preprint arXiv:2402.14989.

---

> > > > > ### Author Response · Authors · 2024-12-01
> > > > >
> > > > > We appreciate the reviewer’s further reviewer and insightful comment. Here are our response for your new comments:
> > > > > >Q1. I understand the choice of model configuration intuitively, but lack of theoretical analysis remains a significant gap.
> > > > > Also, current sensitivity analysis is quite limited, because there is only trend in accordance to the patch size. It is understandable to select hyperparameter, but still questionable to its robustness. Is it related to the data characteristic? How can we select the configuration based on theoretical analysis (or other perspective)?
> > > > >
> > > > > A1. t is related to data characteristics to a certain extent. The masking ratio is related to the missing data ratio. For instance, in irregular time series with high levels of missing data, a higher masking ratio can limit the data available for learning. In such cases, a smaller masking ratio might be preferred. Our results also suggest that a 20-30% masking ratio may help retain the irregularity pattern in the PhysioNet, MIMIC-III and USHCN database.
> > > > >
> > > > > For the patch size, it's related to the sampling rate, and signal length. With a higher sampling rate, we might use a longer patch length as the overall duration will be still limited. For longer sequences, there's a trade-off in balancing the number of patches to manage how fine-grained the local information learned is versus the amount of information contained in each patch. These are practical considerations when determining the hyperparameter selection, and we'll include them in the revised version.
> > > > >
> > > > > >Q2. Did you bound the range of $\omega$ and $\alpha$ in the training process? It seems that these parameter can be fluctuated depending on each samples (or even each variables). How can we guarantee the convergence of these learnable parameters and corresponding ITA values?
> > > > >
> > > > > A2. The sinusoidal function $\(\sin(\omega t + \alpha)\)$ converts the original time index $\(t\)$ into a sinusoidal form. The frequency $\(\omega\)$ and bias $\(\alpha\)$ are optimized to find the best mapping parameters, providing a universal rule but not sample-dependent rule for translating irregular time indices into sinusoidal representations. There are no predefined constraints on $\(\omega\)$ and $\(\alpha\)$ because time series data are complex and non-stationary, lacking clear patterns to define specific frequency or bias ranges. Additionally, imposing task-specific constraints could limit the model's ability to generalize across different patterns. Although $\(\omega\)$ and $\(\alpha\)$ are not explicitly bounded, they are implicitly controlled through optimization. The sine function outputs values within $\([-1, 1]\)$, ensuring stability and preventing $\(\omega\)$ and $\(\alpha\)$ from diverging during training.
> > > > >
> > > > > >Q3.1 For the comparison with, mTAND in Table 1, it seems the results are different from the original paper [1]. Could you elaborate the reason?
> > > > >
> > > > > A3.1 The performance difference in mTAND results in Table 1 compared to the original paper [1] is due to the use of different evaluation metrics. We use Root Mean Squared Error (RMSE) while mTAND uses MSE. When evaluated with the same metric as mTAND, our performance is consistent with reported performance in [1]. RMSE was chosen because it amplifies performance differences, making it ideal for highlighting relative improvements among methods.
> > > > >
> > > > > >Q3.2 For instance, I think the error should be decreased when the observation ratio is increased, but the results in the manuscript are opposite.
> > > > >
> > > > > A3.2 Our trend aligns with the trend reported in mTAND, where an increasing ratio in observations does not necessarily result in improved performance. While we agree that this holds true for regular time series, data sparsity in irregular time series and model complexity can complicate the observations, potentially leading to different outcomes. For example, if we have more data in the observation training set, it will be more highly
> > > > > likely to bias into a certain pattern, if the remaining data shows different distribution, it’s more difficult to impute such distribution.
> > > > >
> > > > > >Q3.3 Furthermore, recent work on Neural ODE/SDE shows better performance in the same task [2].
> > > > >
> > > > > A 3.3 Thanks for pointing out the reference, similar to mTAND, this paper also uses the MSE metrics, and if we convert to the same metric, we still show better performance compared to this method. E.g., at 90% observation range, we achieved 4.87 x 10-2, while the paper presented 5.58 x 10 -2. We will also add this paper into the related work section in the revised version.

---

### Author Response · Authors · 2024-12-02
**Genearl Response**

Dear AC and Reviewers:

We sincerely thank the reviewers for their insightful comments and discussions, which have significantly helped us to address unclear aspects and improve the quality of our paper. Below is a summary of the questions and responses following the reviewer's subsequent comments:

- **Performance Difference** For *Reviewer X6xD*, *Reviewer oiGX* who raised concerns about performance mismatches between our results and the baseline-reported results, we have clarified that the discrepancy is mainly due to the different metrics used. We use RMSE, whereas the references use MSE. When converting to the same evaluation metric of MSE, our proposed method consistently outperforms the referenced baseline systems, e.g., with relative improvements of 12.7% for interpolating 10% of missing data. We chose to use RMSE to better highlight the performance differences among models. Notably, we have ensured a fair comparison by reproducing the baselines according to their official guidelines and maintaining consistent settings, such as using the same number of variables.

- **Baselines** For *Reviewer hwL2*, who suggested a more comprehensive comparison with GAN- and diffusion-based models, we have carefully reviewed the current literature and identified only one relevant study, whose performance is also surpassed by one of our baseline methods. Additionally, there is no strong evidence suggesting that diffusion-based methods are more effective and suitable for irregular time series regression tasks [1]. More importantly, our study focuses on designing SSL frameworks that effectively learn from irregular time series data and generalise to various unseen domains. We followed the standard evaluation protocols of pre-training, fine-tuning, and evaluation, and included state-of-the-art baselines and datasets, consistent with existing work in this area [2,3]. Therefore, comparisons to general diffusion-based methods are outside the scope of this study.

- **Methods**
    - **Multivarite Dependency:** For *Reviewer oiGX* who raise the question of why our channel-independent approach outperforms methods that capture channel correlations, we have provided both the experimental results and conceptual reasoning. Our experiments addressed the reviewer's concern regarding the large dataset size, confirming that it is not the reason for the benefits of our designed channel independence. We have clarified the two reasons as follows: i)Multivariate irregular time series often exhibit distinct sampling rates and dynamic missing patterns across channels, which existing models like Crossformer, designed for regular time series, fail to handle effectively. Learning each variable independently allows our model to bypass these limitations, leading to superior performance. and ii) To achieve generalization across unseen datasets with varying irregularities and channel configurations, our approach avoids pre-training on fixed channel dependencies. For instance, capturing dependencies in 12-lead ECG data would hinder transferability to 2-channel PPG data due to differences in channel types and patterns. Channel-independent learning ensures robust generalization across diverse datasets.

    - **Theoretical Analysis:** For *Reviewer X6xD* who suggested theoretical analysis of parameter configuration, we have provided the practical insights of the parameters constraints. Our paper primarily focuses on empirical analysis supported by comprehensive experiments, and we will clarify the rationale for parameter selection based on real-world dataset characteristics.

Finally, we want to emphasize that the novelty and objective of our work lie in designing a self-supervised learning framework that generalizes across unseen domains with varying irregular patterns, ratios, and numbers of signal channels. This is the first study aimed at transferring a pre-trained model for irregular time series to diverse domains with different data types and channel configurations. Current studies often focus on task-specific models or pre-training and testing on the same dataset due to challenges like data sparsity and unpredictable missing patterns. In contrast, our work presents a promising SSL framework that effectively captures irregular data patterns and transfers them to unseen domains.

Our specific design consists of multiple modules, including an irregular attention module and a temporal dependencies module, within the mask and reconstruction pretraining task. This design effectively captures the intrinsic characteristics of irregular time series. Additionally, we employ a channel-independent training approach to enhance generalization capability across unseen domains.

---

> ### Author Response · Authors · 2024-12-02
>
> References:
>
> [1]Yang, Yiyuan et al. “A Survey on Diffusion Models for Time Series and Spatio-Temporal Data.” ArXiv abs/2404.18886 (2024): n. pag.
>
> [2]Zhang, Weijiao et al. “Irregular Multivariate Time Series Forecasting: A Transformable Patching Graph Neural Networks Approach.” International Conference on Machine Learning (2024).
>
> [3]Chowdhury, Ranak Roy et al. “PrimeNet: Pre-training for Irregular Multivariate Time Series.” AAAI Conference on Artificial Intelligence (2023).

---

### Meta-Review · Area_Chair_Jg4h · 2024-12-21

**Metareview:**

The paper presents a new method for irregular time series regression using masking and reconstruction. The reviewers agreed that the paper is well-motivated, the method is technically sound and the experiments contain a variety of datasets.

However, the paper also suffers from drawbacks including the novelty of the method (a reviewer pointed to T-PatchGNN, another to GAN-based methods as being similar), and several other baselines were recommended (such as diffusion models). While the authors conducted some experiments, they did not test against everything the reviewers recommended.

Overall, while there might be merits to the paper, I cannot recommend acceptance in its current form.

**Additional Comments On Reviewer Discussion:**

The main issues raised are included in the meta-review (novelty and comparisons to baselines).
The reviewers participated in the discussion and ultimately 3 of the 4 decided to reject the paper.

---

### Decision · Program_Chairs · 2025-01-22

Reject